# SHOOT FIRST, ASK QUESTIONS LATER?
# BUILDING RATIONAL AGENTS THAT EXPLORE AND ACT LIKE PEOPLE

**Gabriel Grand**[1,2*]    **Valerio Pepe**[3*]    **Joshua B. Tenenbaum**[1,2]    **Jacob Andreas**[1]
[1]MIT CSAIL    [2]MIT Brain and Cognitive Sciences    [3]Harvard SEAS
*Equal contribution*

## ABSTRACT

Many emerging applications of AI—from scientific discovery to medical diagnosis—require agents to seek information strategically: forming hypotheses, asking targeted questions, and making decisions under uncertainty. In high-stakes settings with limited resources, do language models (LMs) behave like rational agents? Drawing on insights from human cognition, we develop methods to evaluate and enhance agentic information-seeking. First, we introduce a decision-oriented dialogue task called *Collaborative Battleship*, in which a *Captain* must balance exploration (asking questions) and action (taking shots), while a *Spotter* must supply accurate, contextually-grounded answers. Compared to human players (*N=42*), we find that many LM agents struggle to ask informative questions, produce accurate answers, and identify high-utility actions. To address these gaps, we develop novel Monte Carlo inference strategies for LMs inspired by Bayesian Experimental Design (BED). For Spotter agents, our approach boosts accuracy by up to 14.7% absolute over LM-only baselines; for Captain agents, it raises expected information gain (EIG) by up to 0.227 bits (94.2% of the achievable noise ceiling). Combined, these components yield sharper targeting (+0.303–0.374 F1), and enable weaker LMs, such as Llama-4-Scout, to outperform both humans (8% $\rightarrow$ 82% win rate) and frontier models (0% $\rightarrow$ 67% win rate vs. GPT-5) at $\approx$1% of GPT-5's cost. We replicate these findings on *Guess Who?*, where our methods significantly boost accuracy (+28.3–42.4 p.p.), demonstrating their general applicability for building information-seeking agents.

 gabegrand.github.io/battleship

## 1 INTRODUCTION

Language models (LMs) are rapidly evolving from chat-based assistants into fully-fledged agents that interact with the world. Some of the most exciting applications of agents—conducting scientific experiments, conjecturing new mathematical theorems, or discovering novel drugs (Gottweis et al., 2025; Lu et al., 2024; Poesia et al., 2024; Schmidgall et al., 2025)—involve seeking "hits" in combinatorially vast hypothesis spaces. Traditional accounts of information-seeking assume agents are capable of various rational, probabilistic inferences; e.g., inferring belief states, reasoning about uncertainty, and navigating explore/exploit tradeoffs (Anderson, 1990; Auer et al., 2002; Lindley, 1956; MacKay, 1992; Sutton & Barto, 2018). To what extent can current LMs, which are typically optimized to *answer* users' queries (Ouyang et al., 2022; Rafailov et al., 2023), instead *ask good questions* for themselves? And what strategies can we use to improve their information-seeking abilities at inference time?

In this work, we aim to both evaluate and improve the ability of frontier models to ask goal-directed questions and take actions in a dynamic environment. Our setting is an adaptation of the classic board game *Battleship* where players may ask *natural language questions* to gain information about hidden ships. We further extend this paradigm, which was originally developed to study human question-asking (Rothe et al., 2017; 2018; 2019), into a two-player dialogue and decision-making task. We conduct experiments with both human-human and agent-agent pairings, comparing the strategies that LMs employ against both human behavior and idealized resource rational strategies that combine LMs with Bayesian inference techniques.

Figure 1: **Illustration of our *Collaborative Battleship* game.** On each turn, the *Captain* must choose whether to gather information (ask a question) or take action (shoot at a tile). The *Spotter* sees the full board, but can only provide yes/no answers. Each role requires well-defined forms of internal reasoning (thought bubbles), which we implement as Monte Carlo inference over an approximate hypothesis space. This framework allows us to compare both humans and LM agents against idealized Bayesian strategies in a controlled setting.

Our Battleship task tests several distinct cognitive capabilities: (1) *Asking informative questions* that effectively reduce uncertainty; (2) *Providing accurate answers* that are grounded in both the current observation state and the dialogue context; (3) *Taking strategic actions* that leverage available information; (4) *Navigating explore/exploit tradeoffs* in order to balance information-gathering with goal-directed behavior. The minimalistic environment, which shares elements of other challenging text- and grid-based evaluations (Chollet et al., 2024; Guertler et al., 2025; Jansen et al., 2024; Ke et al., 2024; Wang et al., 2022; Yao et al., 2025), provides an ideal testbed for studying Bayesian experimental design (BED; Chaloner & Verdinelli, 1995; Lindley, 1956; Rainforth et al., 2023) in complex state spaces. In particular, questions in Battleship are directly translatable to Python programs; executing these against a sampled "hypothesis space" of game states to compute their expected information gain (EIG) provides a robust way to compare the utility of human and model-generated questions.

As the foundation of our study, we collect 126 full human-human game trajectories (*N*=42 participants), capturing both dialogue and actions. Our BATTLESHIPQA dataset provides two complementary evaluation settings: **SpotterQA**, which tests grounded answering on 931 gold yes/no questions with expert annotations for answers and various question features, and **CaptainQA**, which tests full strategic gameplay under limited questions and shots.

Evaluating current LMs on these benchmarks reveals a wide spectrum of capability. Weaker models like Llama-4-Scout only marginally exceed random baselines on both SpotterQA (62.2% accuracy) and CaptainQA (68.8% win rate vs. random firing), while frontier reasoning models like GPT-5 match or exceed average human performance. On the answering side, we find that Python code generation substantially improves grounding: across 15 LMs, it boosts SpotterQA accuracy by 14.7% over direct answering and chain-of-thought baselines, with especially large gains for models such as GPT-4.1 (75.2% → 90.9%) and Claude 4 Opus (86.8% → 94.4%). On the question-asking side, we introduce a simple Bayesian sampling method that substantially improves question quality, raising mean per-question expected information gain (EIG) on CaptainQA by up to 0.227 bits—94.2% of the information-theoretic ceiling. We also observe that many models tend to ask redundant questions that have zero information gain (e.g., Llama-4-Scout: 18.5% of questions; GPT-4o: 14.6%); our method effectively eliminates these cases.

In total, we introduce three rational strategies for question-asking (Bayes-Q), move selection (Bayes-M), and decision-making (Bayes-D). When combined, these strategies yield substantial improvements in overall game performance, as measured by targeting accuracy (+0.397 F1 for Llama-4-Scout, +0.332 F1 for GPT-4o). Remarkably, these Bayesian enhancements enable even weak LMs to reach superhuman-level performance, with win rates of 81–82% against humans and 67% against GPT-5, all while maintaining substantial cost savings (99.7× for Llama-4-Scout and 2.8× for GPT-4o relative to GPT-5). Finally, we replicate our experiments on the *Guess Who?* task from TextArena (Guertler et al., 2025) and observe similarly significant gains (+42.4 p.p. for Llama-4-Scout and +28.3 p.p. for GPT-4o), demonstrating that our approach generalizes beyond the Battleship domain.

In sum, our work provides both practical and theoretical contributions. Concretely, we introduce a reusable evaluation harness for studying agentic information-seeking, and curate a novel multimodal dataset, BATTLESHIPQA, that captures rich pragmatic phenomena in grounded dialogue. Conceptually, we formalize several Bayesian-inspired inference-time strategies that can be applied to other discovery settings to build rational information-seeking agents.

## 2 THE BATTLESHIP GAME

Our environment draws inspiration from the cognitive science literature, where Battleship-like tasks have previously been utilized to study human information-seeking behavior. In prior work, single-player participants viewed partially-revealed game boards and decided what tiles to reveal (Gureckis & Markant, 2009; Markant & Gureckis, 2012; 2014) or what questions to ask (Rothe et al., 2017; 2018; 2019). Here, we adapt this paradigm to study both humans and language model agents. Our *Collaborative Battleship* game (Fig. 1) is played by two players: a partially-informed *Captain* who must balance exploration (asking questions) and exploitation (taking shots), and a fully-informed *Spotter* who must provide accurate answers that are grounded in both the game state and the ongoing dialogue. Further details about the game rules, interface, and data collection are provided in §A.

Our work extends the *Battleship* paradigm along several key dimensions. (1) *Full multi-turn games:* Prior work was limited to static "snapshots" of game states; here, we simulate full game trajectories. (2) *Dialogue setting:* To enable multi-turn play, we introduce a second player (the Spotter), whose role is necessary to provide real-time answers to the Captain's questions. (3) *Python programs:* In prior work, questions were represented as programs in a hand-engineered domain-specific language (DSL), with translation performed either manually (Rothe et al., 2017) or using LMs (Grand et al., 2024). Here, we represent questions with Python programs, which are both more expressive and easier for LMs to generate. (4) *Information bottleneck:* In prior work, players could in principle ask questions like, "What are the coordinates of all the ships?". To prevent players from asking "game-breaking" questions in order to achieve monetary bonuses (§4.1), we deliberately restrict the Spotter to "Yes" or "No" answers. While less open-ended, such information bottlenecks—also found in games like Twenty Questions, Guess Who, Mastermind, and the original Battleship—ensure strategic balance and enable the study of explore/exploit decision-making.

## 3 FORMAL FRAMEWORK: BAYESIAN EXPERIMENTAL DESIGN

We cast question selection in our Battleship variant as Bayesian Experimental Design (BED): on each turn we choose to explore—by asking a yes/no question to gain information—or to act—by firing at a hidden tile.

- The hidden board is a random variable $S \in \mathcal{S}$. The observed partial board $x$ induces the feasible set $\mathcal{S}_{\vdash x} = \{ s \in \mathcal{S} : s \text{ is consistent with } x \}$.
- The belief at the start of turn $t$ given history $\mathcal{H}_{1:t}$ is $\pi_t(s) = \Pr(S = s \mid x, \mathcal{H}_{1:t})$. Intuitively, this belief places weight on every board that could still be true given what we have seen so far.
- A natural-language question $q$ translates to a deterministic function $f_q : \mathcal{S} \rightarrow \{0, 1\}$. The true (noise-free) answer is $A_q := f_q(S) \in \{0, 1\}$. For the question asked at turn $t$, we write $A_t := f_{q_t}(S)$.

**Observation Model (Noisy Spotter)** In our setting, the Spotter plays the role of an observation model. Assuming an oracle Spotter, the predictive probability that the true answer will be "yes" is simply

$$p_t := \Pr(A_t = 1 \mid x, \mathcal{H}_{1:t}) = \sum_{s \in \mathcal{S}_{\vdash x}} \pi_t(s)\, \mathbf{1}\{f_{q_t}(s) = 1\}. \tag{1}$$

We expect that the Spotter (human or AI) may occasionally make mistakes, either because they misread the board, misinterpreted the question, or miscommunicated the answer. Accordingly, we model the Spotter (and language-to-code translation) as a *binary symmetric channel* $\mathrm{BSC}(\varepsilon)$ with flip probability $\varepsilon \in [0, \frac{1}{2}]$, yielding a noisy answer $\widetilde{A}_t \in \{0, 1\}$.

**Bayesian Belief Update.** We start with a basic model of the Captain as a Bayesian ideal observer. After asking $q_t$, the Captain updates their belief by reweighting each candidate board $s \in \mathcal{S}_{\vdash x}$ by how compatible it is with the observed answer $\tilde{a}_t \in \{0, 1\}$ under the noise model:

$$\pi_{t+1}(s) \propto \pi_t(s) \left[ (1 - \varepsilon)\, \mathbf{1}\{\tilde{a}_t = f_{q_t}(s)\} + \varepsilon\, \mathbf{1}\{\tilde{a}_t \neq f_{q_t}(s)\} \right]. \tag{2}$$

*Intuition.* Boards that would have produced the observed answer (e.g., $f_{q_t}(s) = \tilde{a}_t$) get *boosted* by a factor $1 - \varepsilon$, while boards that would have said the opposite retain some weight $\varepsilon$ because mistakes

are possible. This gently shifts mass toward boards consistent with what we saw without discarding alternatives outright when $\varepsilon > 0$.

*Sequential Monte Carlo (SMC) Approximation.* Exact sums over $\mathcal{S}_{\vdash x}$ are typically intractable; we therefore maintain a weighted particle approximation $\{(s_j, w_j^{(t)})\}_{j=1}^N$ which is updated via sequential Monte Carlo (SMC; Doucet et al., 2001) with per-turn resampling; see §B for details.

**Expected Information Gain.** To decide which of many possible candidate questions $q_t$ to ask, we adopt a standard information-theoretic approach (MacKay, 1992; Shannon, 1948) that considers the expected information the Captain will gain about the board from its (noisy) answer:

$$\text{EIG}_\varepsilon(q_t \mid x, \mathcal{H}_{1:t}) \;:=\; I(S; \widetilde{A}_t \mid x, \mathcal{H}_{1:t}) \tag{3}$$

For a $\text{BSC}(\varepsilon)$, this admits a closed form in terms of the binary entropy $\text{H}_b$:

$$\boxed{\text{EIG}_\varepsilon(q_t \mid x, \mathcal{H}_{1:t}) = \text{H}_b\big(\varepsilon + (1 - 2\varepsilon)\, p_t\big) \;-\; \text{H}_b(\varepsilon), \quad p_t \text{ as in (1)}} \tag{4}$$

*Intuition.* The first term $\text{H}_b(\varepsilon + (1 - 2\varepsilon)p_t)$ is the uncertainty in the *noisy* answer we expect to hear; the second term $\text{H}_b(\varepsilon)$ is the uncertainty injected by the channel itself. Their difference is how much uncertainty about the board we expect to remove by asking this question. This quantity is maximized when $p_t \approx \frac{1}{2}$, i.e., when under our current belief the question is about as likely to return "yes" as "no".

### 3.1 BAYESIAN STRATEGIES FOR EXPLORATION AND ACTION

We now describe three strategies that leverage the formal framework above to (i) ask questions that maximize expected information gain, (ii) select actions that maximize hit probability, and (iii) decide between asking a question or taking an action at each turn. These strategies are not necessarily globally optimal, but rather *resource rational* in the sense that they use the current belief $\pi_t$ to maximize expected utility at each turn.

**Asking questions to maximize information gain ($Q_{\textbf{Bayes}}$).** To maximize expected information gain, a simple strategy is to sample a set of candidate questions $\mathcal{Q}$ (e.g., from an LM) and select the one with highest $\text{EIG}_\varepsilon$:

$$q_t^\star \in \underset{q \in \mathcal{Q}}{\arg\max} \; \text{EIG}_\varepsilon(q \mid x, \mathcal{H}_{1:t}), \tag{5}$$

**Selecting moves to maximize hit probability ($M_{\textbf{Bayes}}$).** For a candidate tile $u$ (restricted to unrevealed locations), define the probability of a hit under the current belief and the corresponding myopic maximum a posteriori (MAP) action:

$$u_t^\star \in \underset{u \text{ unrevealed}}{\arg\max} \; p_t^{\text{hit}}\big(u \mid x, \mathcal{H}_{1:t}\big), \quad p_t^{\text{hit}}\big(u \mid x, \mathcal{H}_{1:t}\big) := \sum_{s \in \mathcal{S}_{\vdash x}} \pi_t(s)\mathbf{1}\{u \text{ contains ship in } s\} \tag{6}$$

**Making rational decisions via one-step lookahead ($D_{\textbf{Bayes}}$).** There are many possible strategies for deciding whether to ask a question or take a shot; here, we describe a simple planning-based approach that uses a discounted one-step lookahead to estimate the value of asking a question vs. taking a shot.

*Expected post-question hit probability.* For a candidate question $q_t$, we can estimate the next-turn MAP hit probability by marginalizing over possible answers $\tilde{a}$

$$\widehat{p_{t+1}^{\text{hit}}}\big(u \mid x, \mathcal{H}_{1:t}, q_t\big) := \sum_{\tilde{a} \in \{0,1\}} \Pr(\widetilde{A}_t = \tilde{a} \mid x, \mathcal{H}_{1:t}, q)\, p_{t+1}^{\text{hit}}\big(u \mid x, \mathcal{H}_{1:t}, \tilde{a}\big). \tag{7}$$

Here $\Pr(\widetilde{A}_t = 1 \mid \cdot) = \varepsilon + (1 - 2\varepsilon)\, p_t$ with $p_t$ as in (1), and the inner hit probability is evaluated under the updated belief given $\tilde{a}$. Let $\gamma \in [0, 1]$ discount the value of information acquired this turn. The strategy proceeds as follows:

1. Select a *potential* question $q_t^\star$ by maximizing EIG via Eq. (5).
2. Compute the pre- and post-question MAP moves $u_t^\star$ and $u_{t+1}^\star$ and corresponding hit probabilities as in Eqs. (6) and (7).
3. If $\gamma\, \widehat{p_{t+1}^{\text{hit}}}\big(q_t^\star \mid x, \mathcal{H}_{1:t}\big) > p_t^{\text{hit}}\big(u_t^\star \mid x, \mathcal{H}_{1:t}\big)$, ask $q_t^\star$; otherwise, act (shoot) at $u_t^\star$.

*Intuition.* When $\gamma = 1$, the policy asks a question iff the expected one-step improvement in the next-turn MAP hit probability exceeds the current MAP chance; smaller $\gamma$ prefers acting sooner. While optimizing over longer horizons is possible, the problem of belief-space planning is PSPACE-hard (Papadimitriou & Tsitsiklis, 1987); in practice one-step lookahead is simple and effective.

## 4 EXPERIMENTS

We present our experimental evaluation in three parts. First, in §4.1, we describe our human behavioral study and dataset, BATTLESHIPQA, in order to build an empirical account of human-like information seeking. Next, we present a series of experiments that evaluate the ability of LMs to answer and ask questions, leveraging BATTLESHIPQA in two distinct ways. In §4.2, we focus on grounded question-answering, evaluating the performance of LMs in answering questions asked by humans. Finally, in §4.3, we focus on question-asking, evaluating the ability of LMs to ask informative questions and make strategic moves in the full game setting.

### 4.1 EVALUATING HUMAN INFORMATION-SEEKING BEHAVIOR

We conducted a two-player, synchronous behavioral study in which participants played *Collaborative Battleship* (described in §2). Participants (N=42) were recruited from Prolific and randomly partnered with another participant. Each pair alternated between the Captain and Spotter roles over 6 games, with a total time commitment of 48-60 minutes and average earnings of $12.03-$14.62/hr (including bonuses based on targeting score). In total, we collected data for 18 pre-sampled, 8×8 game boards, allocated evenly across pairs. Further details about the experimental design, implementation, and compensation are provided in §A.2.

**Analysis of human data.** We manually annotated the human data in order to (1) establish a set of gold labels for evaluating model performance and (2) categorize the types of questions being asked. First, we annotated all questions with a "gold answer" label and filtered out questions for which there was inter-annotator disagreement. After this process, we obtained a gold dataset of 931 questions, establishing a human accuracy baseline of 92.5%.

Next, we categorized questions into two groups based on whether they could be answered solely from the true board $S$ (*simple*) or required additional context from the game history $\mathcal{H}_{1:t}$ (*complex*). Within the "complex" category, we futher annotated for various linguistic and pragmatic phenomena, such as *discourse-dependence* (e.g., "Is it longer than 3 tiles?" where "it" references a specific ship), *state-dependence* (e.g., "Should I keep firing up?"), *vagueness* (e.g., "Is there a ship near the center?") and *ambiguity* (i.e., multiple possible interpretations). Further definitions and details are given in §A.3; illustrated examples are provided in §A.4.

### 4.2 SPOTTERQA: MODELING QUESTION-ANSWERING

We begin our modeling experiments with a focused evaluation of models' performance in the Spotter role, since providing accurate answers is a crucial signal of models' ability to reason about the game state. In particular, we are interested in two answering strategies that we hypothesize should improve

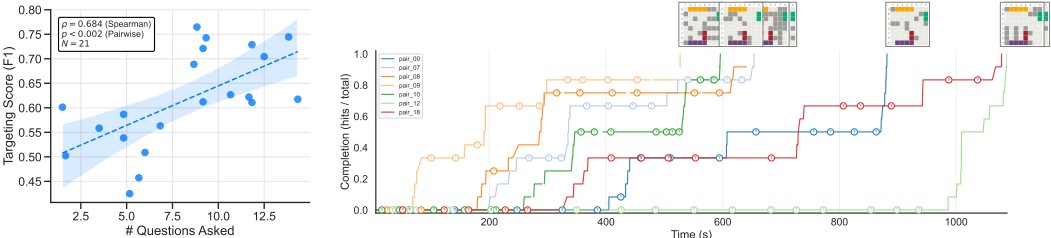

(a) Mean targeting score (F1) vs. # questions asked for each human pair.

(b) Timeline of multiple human-human games of Battleship for a single board (B02). Each ⑦ represents a question asked; colors indicate different players.

Figure 2: **Human results highlights.** (a) Asking questions correlates with performance ($\rho = 0.684, p < 0.002$); (b) Different players demonstrate widely varying explore/exploit strategies; some alternate between asking questions and making moves, while others focus on information-gathering before taking any actions.

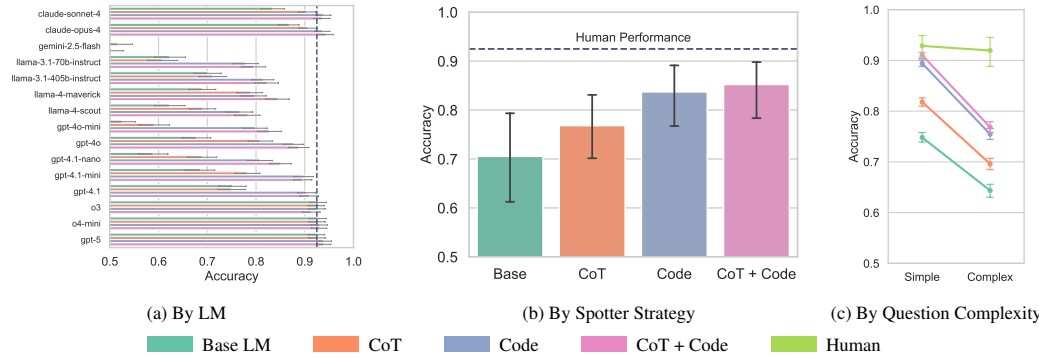

Figure 3: **SpotterQA key results.** (a) Accuracy varies significantly across the 15 LMs tested. (b) Across LMs, code generation improves accuracy over direct answering and CoT baselines. (c) Whereas human accuracy remains roughly consistent, LMs degrade significantly on "complex" questions that require context. Dashed lines indicate mean human accuracy ($\mu = 92.5\%$) w/r/t gold answer labels. Error bars indicate 95% CIs.

performance: chain-of-thought reasoning (**CoT**; Kojima et al., 2022; Nye et al., 2021; Wei et al., 2022) and language-to-code translation (**Code**; Austin et al., 2021; Wong et al., 2023, see §D.1.1 for examples of synthesized programs). We evaluate these two strategies alongside a combined **CoT + Code** condition and a direct-answer **Base LM** condition.

**Evaluation Details** Each item in the SpotterQA benchmark is a tuple $(q_t, A_t, x_t, \mathcal{H}_{1:t})$ containing a question, answer, observed board, and history from the human experiment. Boards are represented as Numpy arrays (prompting details in §C.3).[1] Since questions are independent given the history, sequences can be split up for efficient parallel processing. Accordingly, we are able to evaluate a broad (but non-exhaustive) set of frontier LMs from Anthropic (Anthropic, 2025), Google (Comanici & Team, 2025), Meta (Grattafiori et al., 2024; Meta AI, 2025), and OpenAI (OpenAI, 2024; 2025a;b;c). We query all models via the OpenRouter API with default parameters.

### 4.2.1 RESULTS

Fig. 3 summarizes the results of our SpotterQA evaluation; full details are provided in §D.1. We highlight three key findings:

**Question-answering abilities vary widely across models.** As seen in Fig. 3a, accuracy varies widely across the 15 LMs tested, ranging from near-chance (52.5%; GPT-4o-mini) to 92.8% (o3 mini). Notably, several models (o3, o4-mini, and GPT-5) meet or exceed mean human performance (92.5%). (See Table 3 for full results.)

**Code generation consistently improves answering accuracy.** Across models (Fig. 3b), we find that code improves SpotterQA by 13.2% absolute accuracy points over Base; combining code generation with chain-of-thought (CoT + Code) yields even greater 14.7% gains. A two-sided Mann-Whitney U test confirms that these differences are significant ($p < 0.001$; see Fig. 17). Importantly, for many models, code generation substantially closes the gap between LMs and human performance: for instance, Claude 4 Opus improves from 86.8% (Base) to 94.4% (CoT + Code), and GPT-4.1 improves from 75.2% (Base) to 90.9% (CoT + Code). These results demonstrate that code generation helps LMs to produce more accurate, grounded answers.

**LMs struggle with context-dependent questions.** As shown in Fig. 3c, whereas human accuracy remains roughly consistent across simple (92.8%) and complex (91.9%) questions, LMs degrade significantly on complex questions (as defined in §4.1). For instance, GPT-4o's accuracy drops from 72.8% on simple questions to 60.4% on complex ones; similarly, Llama-4-Scout drops from 68.0% to 54.0%. Code generation partially mitigates this gap, but even the best model (o3) still falls short of human performance on complex questions (87.4% vs. 91.9%). These results suggest that current LMs still struggle with pragmatic reasoning and context-dependent meanings.

---

[1]Prior evaluations find that vision language models (VLMs) struggle with spatial reasoning on 2D grids, including Battleship boards (Grand et al., 2024). Accordingly, we use serialized representations, following the predominant approach on ARC-like tasks (Akyürek et al., 2024; Li et al., 2024; Wang et al., 2024).

### 4.3 CAPTAINQA: MODELING QUESTION-ASKING

In this section, we evaluate the ability of LMs to play the Captain role in the full game setting. We compare a variety of Captain strategies that differ in how they select moves, ask questions, and decide whether to ask or shoot (as formalized in §3.1). A summary is provided in Table 1.

| Captain | Decision (D) | Question (Q) | Move (M) |
|---|---|---|---|
| Random | Move | – | $i, j \sim \mathrm{Unif}(\{1, \ldots, 8\})$ |
| Greedy | Move | – | $\arg\max_{i,j} \pi_t(\cdot \mid x)$ |
| LM | $p_{\mathrm{LM}}(\cdot \mid x, \mathcal{H}_{1:t})$ | $p_{\mathrm{LM}}(\cdot \mid x, \mathcal{H}_{1:t})$ | $p_{\mathrm{LM}}(\cdot \mid x, \mathcal{H}_{1:t})$ |
| + Bayes-Q | ⋮ | $\arg\max_{q \in \mathcal{Q}} \mathrm{EIG}_\varepsilon(q \mid x, \mathcal{H}_{1:t})$ | ⋮ |
| + Bayes-M | ⋮ | $p_{\mathrm{LM}}(\cdot \mid x, \mathcal{H}_{1:t})$ | $\arg\max_{i,j} \pi_t(\cdot \mid x, \mathcal{H}_{1:t})$ |
| + Bayes-QM | ⋮ | $Q_{\mathrm{Bayes}}$ | $M_{\mathrm{Bayes}}$ |
| + Bayes-QMD | $p^{\mathrm{hit}}(\pi_t) > \gamma p^{\mathrm{hit}}(\pi_{t+1} \mid q_t, \widetilde{A}_t)$ | ⋮ | ⋮ |

Table 1: Summary of Captain strategies. *Random* and *Greedy* are move-only baselines; *LM* is a pure language model, which the *Bayes* strategies build upon. Triple-dots indicates inheritance from the row above.

**Experimental Details** We evaluate each Captain strategy over 54 games (the 18 pre-sampled boards from the human experiment; §4.1, with 3 random seeds per board). Each game enforces the same constraints as the human setting: a maximum of 15 questions and 40 moves. To control for answer quality, we fix the Spotter to GPT-5 (CoT + Code) for all strategies and set $\varepsilon = 0.1$, calibrated from GPT-5's SpotterQA accuracy (§4.2). Because each turn involves multiple API queries which must be made sequentially, we restrict CaptainQA evaluation to three representative LMs: Llama-4-Scout (small, non-reasoning), GPT-4o (large, non-reasoning), and GPT-5 (large, reasoning-capable). We evaluate every combination of LM and Captain strategy, with the exception of Bayes-QMD, which we were not able to evaluate with GPT-5 due to cost (see Table 5 for a breakdown). For $D_{\mathrm{Bayes}}$, we set $\gamma = 0.95$ to encode a minor discount for future action. For strategies that use LMs, the prompt includes the current board state and the full game history (see §C.2 for prompting details).

**Captain Evaluation Metrics** We evaluate Captains using five primary metrics that capture different aspects of performance:

- **Targeting Score (F1)**: Overall metric of ship-sinking performance, balancing both precision and recall. This metric treats the board as a binary classification task, where ship tiles correspond to the positive class and water tiles correspond to the negative class.
- **Move Count**: The average number of shots taken per game; equivalently, the average game length (max 40 moves; questions do not count towards the move count).
- **Questions Asked**: The average number of questions asked per game (max 15).
- **Win Rate**: A pairwise metric based on board-matched performance in simulated head-to-head play. For each pair of Captains on the same board, the winner is the one that sinks all ships in the fewest moves, with tiebreaking based on targeting score (F1). Since Captains do not play each other directly, this comparison is computed post-hoc over all pairs of games, averaging over boards.
- **EIG**: The average expected information gain of questions asked. We use $\varepsilon = 0.1$, meaning that the maximum possible EIG is $1 - \mathrm{H}_b(0.1) = 0.531$ bits.

#### 4.3.1 RESULTS

Fig. 4 summarizes the results of our CaptainQA evaluation; full details are provided in §D.2. We highlight four key findings:

**Incorporating Bayesian strategies brings weaker models up to super-human performance.** As seen in Fig. 4a, the LM-only strategy shows a wide range of performance, with Llama-4-Scout achieving only 0.367 F1, while GPT-5 reaches 0.716. However, adding Bayesian question, move, and decision strategies (+Bayes-QMD) significantly improves performance across all models (e.g., Llama-4-Scout jumps $0.367 \to 0.764$ F1, GPT-4o $0.450 \to 0.782$ F1). Remarkably, with the full Bayesian model, both Llama-4-Scout and GPT-4o outperform both humans (0.82–0.83 win rate vs. humans) as well as the strongest LM (0.67 win rate vs. GPT-5), suggesting that Bayesian strategies can compensate for weaker LM capabilities (see win rates in Fig. 18). The LMs equipped with this full Bayesian model also outperform GPT-5 at a fraction of its cost ($\approx 1\%$ of GPT-5's cost for Llama-4-Scout, and $\approx 35\%$ for GPT-4o, see Table 5) Notably, GPT-5 itself does not significantly benefit from Bayesian question or move selection, indicating that it may already be employing effective versions of these strategies internally.

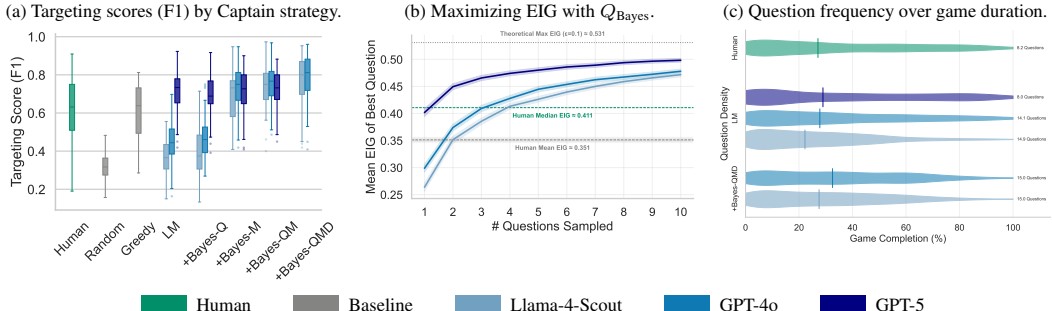

Figure 4: **CaptainQA key results.** (a) Incorporating Bayesian strategies for questions ($Q_{\text{Bayes}}$), moves ($M_{\text{Bayes}}$), and decisions ($D_{\text{Bayes}}$) brings weaker LMs from near-random performance to super-human levels. (b) Sampling up to 10 questions with $Q_{\text{Bayes}}$ yields higher EIG. (c) $D_{\text{Bayes}}$ helps Llama-4-Scout and GPT-4o to spread out questions over time, more closely matching the behavior of humans and GPT-5.

**Inference scaling yields more informative questions for all models.** As shown in Fig. 4b, EIG scales with the number of candidate questions sampled under the $Q_{\text{Bayes}}$ strategy. Across models, we see EIG improvements of up to 0.227 bits per question (up to 94.2% of the info-theoretic ceiling). Additionally, $Q_{\text{Bayes}}$ significantly reduces the proportion of *redundant* questions (EIG = 0; Table 4) asked by Llama-4-Scout (18.5% → 0.2%) and GPT-4o (14.6% → 1.2%). (We find that both humans and GPT-5 rarely ask redundant questions.) These findings illustrate how LMs—including models like GPT-5 that already perform extensive test-time reasoning—can further benefit from inference scaling techniques designed to improve question quality.

**Asking high-EIG questions is not sufficient to guarantee strong game performance.** While $Q_{\text{Bayes}}$ consistently improves EIG, this does not consistently translate to improved game performance. For instance, both Llama-4-Scout and GPT-4o see significant EIG gains with $Q_{\text{Bayes}}$, but their targeting scores improve only marginally (+0.021–0.026 F1). This suggests that weaker models may not be able to effectively leverage information into accurate moves. In contrast, $M_{\text{Bayes}}$, which explicitly marginalizes over the implications of each question to compute $\pi_t$, provides a more reliable mechanism for doing so.

**Skilled players ask *some* questions first, *but not all*.** As seen in Fig. 4c, both humans and GPT-5 tend to ask several questions early in the game, but also reserve some questions for later turns. In contrast, weaker players (e.g., Llama-4-Scout) tend to front-load all 15 questions—this myopic behavior is mitigated by introducing one-step lookahead ($D_{\text{Bayes}}$), which leads to a more balanced approach. Interestingly, stronger players (humans, GPT-5) ask *fewer* questions overall (8.0–8.2 vs. 14.1–14.9), indicating that they both gain more information per question (higher EIG; Table 4) and make more effective use of that information in their moves (higher F1; Fig. 4a).

# 5 GENERALIZED INFORMATION-SEEKING GAMES

We extend our Bayesian strategies to a general family of information-seeking games from TextArena (Guertler et al., 2025). Here, we present results from the board game "*Guess Who?*" (see §E for full details). This game provides a distinct testbed from *Battleship*, as it involves richer, object-relational semantics and requires more complex reasoning about entities and attributes (e.g., age, clothing, facial hair, etc.). As shown in Fig. 5, both GPT-4o and Llama-4-Scout's success rates improve significantly with $QM_{\text{Bayes}}$ (GPT-4o: 0.617 → 0.900; Llama-4-Scout: 0.300 → 0.724). These results suggest our framework successfully generalizes to information-seeking environments with combinatorial hypothesis spaces.

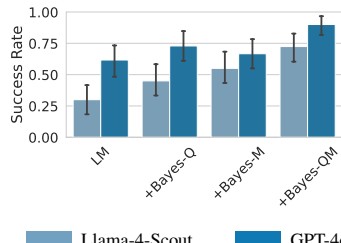

Figure 5: *Guess Who?* Our Bayesian strategies yield consistent improvements, replicating the core findings from *Battleship*.

# 6 RELATED WORK, DISCUSSION, AND CONCLUSION

**Expressing queries and hypotheses with programs.** Several notable prior works use language-to-code to support efficient probabilistic inferences (Ellis, 2023; Li et al., 2024; Piriyakulkij et al., 2024; Wang et al., 2024; Wong et al., 2023). Our work integrates hypothesis-driven reasoning with other recent approaches for planning and rational agent behavior (Chiu et al., 2023; Curtis et al., 2025; Ying et al., 2024; 2025).

**Eliciting user preferences and clarifying ambiguity.** Asking informative questions is critical for many user-facing applications of LMs; recent approaches consider structured prompting (Li et al., 2023), entropy reduction (GX-Chen et al., 2025; Hu et al., 2024; Mazzaccara et al., 2024; Piriyakulkij et al., 2023; Rao & Daumé III, 2018; Yu et al., 2020; Zhang & Choi, 2023), Bayesian inference (Handa et al., 2024; Qiu et al., 2025), and constraint satisfaction (Li et al., 2025).

**Human information-seeking and resource rationality.** Our work is broadly informed by "resource rational" accounts that describe how human behavior is shaped by cognitive constraints (Anderson, 1990; Chater & Oaksford, 1999; Icard, 2025; Leider et al., 2025; Lieder & Griffiths, 2020). In information-seeking tasks, both children and adults greedy heuristics (Markant et al., 2016; Meder et al., 2019; Ruggeri et al., 2016), consider only a few hypotheses at a time (Klayman & Ha, 1989; Vul et al., 2014), and prefer queries that yield easily interpretable information (Cheyette et al., 2023). These ideas motivate our modeling approach, which emphasizes greedy, sample-based strategies over exact planning and inference.

## 6.1 LIMITATIONS AND FUTURE WORK

*Collaborative Battleship* gives rise to many rich pragmatic behaviors. While discussed in §§ A.3 and A.4, in general, these are not explicitly modeled; incorporating techniques based on the rational speech acts (RSA) framework (Frank & Goodman, 2012; Hawkins et al., 2017; 2023) could yield agents capable of more sophisticated pragmatic reasoning. In particular, people are highly sensitive to the reliability of information (Harris & Corriveau, 2011; Sperber et al., 2010), as reflected in interactions between human players (e.g., Figs. 14 and 15). In place of a fixed $\varepsilon$, a more robust approach would be to *infer* $\varepsilon$ to account for the differences in reliability across individual Spotters. Analogously, in scientific settings, agents may need to adapt to different levels of aleatoric uncertainty (Hüllermeier & Waegeman, 2021).

Our Bayesian strategies rely on the ability to efficiently draw conditional samples $s \sim p(s \mid x, \mathcal{H})$ from a generative "world model." While implementable by hand in a domain like Battleship, in more general settings, we may wish to *learn* a generative model of world states represented as code (e.g., via model synthesis architectures (MSA); Wong et al., 2023; 2025) or images; (e.g., via VAEs or diffusion; Alonso et al., 2024; Ha & Schmidhuber, 2018). Finally, building agents that collaborate effectively with people is increasingly important (Boiko et al., 2023; Noti et al., 2025); *Collaborative Battleship* provides an ideal setting for studying human-agent interactions.

## 6.2 CONCLUSION

In this work, we presented a comparative evaluation of human and agent information-seeking. We introduced a cognitively-inspired *Collaborative Battleship* task designed to replicate the core components of Bayesian Experimental Design (BED), including forming hypotheses about latent variables, asking and answering questions, and leveraging context-dependent information to inform actions. Our behavioral study provides a rich, multimodal dataset of human interactions, BATTLESHIPQA, enabling systematic comparison with model performance.

Our results highlight both progress and gaps: while smaller models like Llama-4-Scout and GPT-4o struggle to explore and act coherently, larger reasoning models like GPT-5 approach or even surpass human-level performance at steep resource costs. Notably, humans themselves are not Bayes-optimal reasoners; in general they do not ask maximally-informative questions or make exhaustive use of all available resources. Nevertheless, people are remarkably good at asking useful questions, providing contextually-grounded answers, and making well-informed guesses. Prioritizing strategies that are resource rational is therefore essential if we want to build agents that scale beyond benchmarks to collaborate with people on real-world problems.

AUTHOR CONTRIBUTIONS

**Gabriel Grand\*** Research conceptualization, narrative development, modeling (design and implementation), human experiments (design and supervision), analysis (Captain and Spotter experiments), figure-making, writing, math and formalization.

**Valerio Pepe\*** Research conceptualization, narrative development, modeling (design and implementation), human experiments (design, implementation, and execution), analysis (Captain and Spotter experiments), writing, TextArena experiments.

**Joshua B. Tenenbaum** Senior mentorship, research conceptualization, narrative development, human experiments (design and supervision).

**Jacob Andreas** Senior mentorship, research conceptualization, narrative development, human experiments (design and supervision), analysis of results.

*\*Equal contribution, co-first authorship.*

ACKNOWLEDGMENTS

We thank Brenden Lake, Tomer Ullman, Robert Hawkins, Judy Fan, Lionel Wong, Tan Zhi-Xuan, Belinda Li, Alexis Ross, Sam Cheyette, Guy Davidson, and Graham Todd for helpful discussions and feedback.

The authors gratefully acknowledge support from the Siegel Family Quest for Intelligence, the MIT-IBM Watson AI Lab, the FinTechAI@CSAIL initiative, the Intel Corporation, AFOSR, DARPA, ONR, and the National Science Foundation (NSF) under grants CCF-2217064 and IIS-2238240. G.G. is additionally supported by a NSF Graduate Research Fellowship under Grant No. 2141064. J.B.T. is additionally supported by AFOSR, ONR Science of AI, and Siegel Family Endowment. J.A. is additionally supported by a Sloan Research Fellowship. Any opinions, findings, and conclusions or recommendations expressed in this material are those of the author(s) and do not necessarily reflect the views of sponsors.

ETHICS STATEMENT

We acknowledge that our research involves human participants and has been conducted with the utmost consideration for ethical standards. The study protocol was reviewed and approved by the MIT Committee on the Use of Humans as Experimental Subjects (COUHES), which serves as the Institutional Review Board (IRB) at our institution, to ensure compliance with the Ethical Principles and Guidelines for the Protection of Human Subjects of Research, including the Belmont Report principles of respect for persons, beneficence, and justice. All participants provided informed consent, and their privacy was protected throughout the study. We are committed to ensuring that our findings are used responsibly and for the benefit of society.

REPRODUCIBILITY STATEMENT

We strive to make our research reproducible by providing detailed descriptions of our methods, datasets, and experimental procedures. We encourage other researchers to replicate our findings and build upon our work. All code and data used in our experiments, including the full BATTLESHIPQA dataset, are available at `gabegrand.github.io/battleship`.

USE OF AI ASSISTANCE

We acknowledge the use of AI tools in the preparation of this manuscript. Specifically, we utilized language models (LMs) to assist with grammar and style editing, literature review, preparation of figures, general coding assistance, and other tasks. We have reviewed and edited the content to ensure accuracy and clarity, and we take full responsibility for the final version of the manuscript.

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

# Appendix

## Table of Contents

# A  BATTLESHIP

## A.1  EXTENDED GAME DESCRIPTION AND PRIOR WORK

Our environment draws inspiration from the cognitive science literature, where Battleship-like tasks have previously been utilized to study human information-seeking behavior. In prior work, single-player participants viewed partially-revealed game boards and decided what tiles to reveal (Gureckis & Markant, 2009; Markant & Gureckis, 2012; 2014) or what questions to ask (Rothe et al., 2017; 2018; 2019). In particular, Rothe et al. developed a novel approach where individual questions for specific intermediate game states were elicited from participants and translated into programs in a domain-specific language (DSL) in order to compute various formal characteristics, such as the expected information gain (EIG), minimum description length (MDL), and answer type (Boolean, number, color, etc.). A log-linear model was fit to human data in order to determine relative feature importance and to score the likelihood of questions sampled from a generative grammar defined over the DSL.

**Collaborative Battleship** In our *Collaborative Battleship* game (Fig. 1), players alternate between two roles: the Captain must efficiently navigate explore/exploit tradeoffs in order to gain information about the board through questions and actions, while the Spotter must provide grounded answers to (possibly ambiguous) questions given full visibility of the board. Games start from a blank board; at every turn, the Captain chooses whether to (a) reveal a tile by "shooting" or (b) ask a question. When a question is asked, the Spotter—who sees the full board—is prompted to answer it. Captains are given a limited budget of 15 questions and 40 shots, which under mild assumptions is sufficient to sink all ships. The game ends when either all ships are sunk (team win) or the Captain exhausts their shot budget (team loss). Further details about the game rules, interface, and data collection are provided in §A.2.

**Key methods contributions** Relative to prior studies, our work extends the Battleship paradigm in several key dimensions. (1) *Full multi-turn games:* Prior work was limited to one or two-turn snapshots; here, we simulate full games consisting of interleaved sequences of up to 40 moves and 15 questions. (2) *Two-player setting:* In the multi-turn setting, players need to receive real-time answers to their questions; to facilitate this interaction loop, we introduce a second player to act as a "Spotter," detailed above. (3) *Python programs:* In prior work, question-meanings were manually translated into programs. While follow-up work from Grand et al. (2024) proposed using LMs to automate this process, programs were still restricted to a hand-engineered DSL. Here, we represent questions with Python programs, which are both more expressive and easier in practice for LMs to generate. (4) *State space complexity:* Prior work used small $6 \times 6$ boards; we expand to $8 \times 8$ boards, which allow for more ships and position combinations and make question-asking more integral to skilled play.

**Information Bottleneck** There is also one key aspect in which our setting is intentionally more *restricted*. Prior work placed no formal restrictions on questions; in principle, this meant that players could ask questions like, "What are the coordinates of all the ships?"[2] Rather than attempt to detect such questions in real-time, we instead simply restrict the Spotter to answer with "Yes" or "No." As discussed in §3, this rule acts as an *information bottleneck* that limits the channel capacity. While less open-ended than the original setting, this approach is much more directly-enforceable, which is critical in our setting where human players are incentivized with real monetary rewards (§4.1), and where teams can and do form ad-hoc conventions across rounds.

---

[2]To discourage "game-breaking" questions, Rothe et al. (2017) instructed participants to only ask questions that could be answered with a single word. Nevertheless, there are many ways of constructing a nominally "one-word" answer that encodes the game state (e.g., *RedB1B3PurpF4F7...*). Since the language of such encodings is undecidable, we instead opt to enforce a binary information bottleneck.

## A.2 Human Study Details

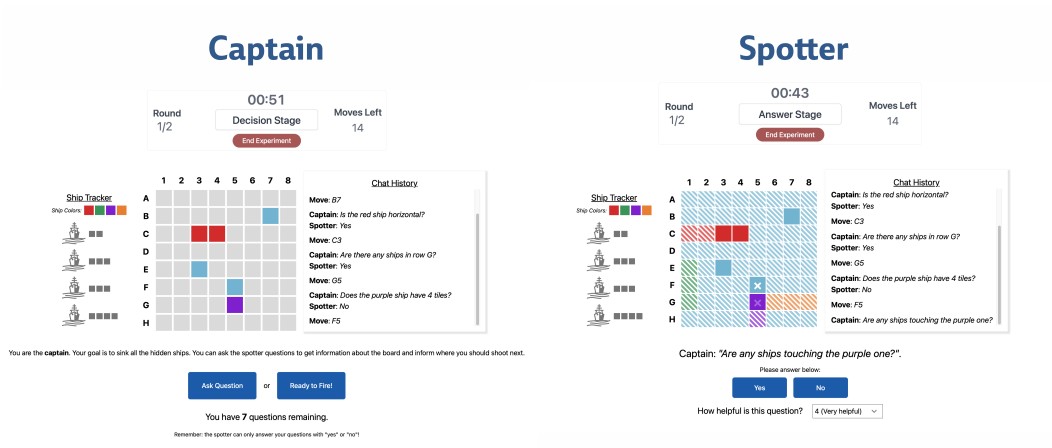

Figure 6: User interface for our *Collaborative Battleship* experiment, which evaluates information-seeking in a two-player synchronous dialogue environment. The Captain must choose between taking actions and asking questions given limited visibility, while the Spotter sees the whole board, but can only respond with Yes/No. In order to play at human-level, an AI agent must incorporate grounded language understanding (to answer questions), reasoning about uncertainty (to ask informative questions), and strategic decision making (to balance explore/exploit trade-offs).

### A.2.1 Study Setup

We recruited N=42 participants from Prolific (www.prolific.com). Participants were required to be located in the United States, fluent in English, and maintain a Prolific approval rate of at least 95%.

The study was conducted with IRB approval in multiple sessions that occurred between November 26, 2024 and January 31, 2025. We conducted the study in a synchronous, two-player format using the Empirica platform (Almaatouq et al., 2021). Participants were matched via Empirica's built-in matchmaking system, which pairs participants as they arrive. Participants were required to use a desktop or laptop computer with a modern web browser (Chrome, Firefox, Edge, or Safari). Mobile devices were not supported.

The human study included a tutorial to familiarize participants with the rules of the game; we did not assume prior experience with Battleship. However, we did set specific criteria on Prolific to exclude participants who had previously completed any prior study (e.g., pilot studies) from our group that specifically used the Collaborative Battleship protocol.

Median time to complete the study was 48–60 minutes, depending on the session. Participants earned a base rate of $8.00 for completing the study, with performance-contingent bonuses of $0.20 per hit and -$0.10 per miss (with a bonus floor of $0 such that base pay was always guaranteed). In total, average per-participant earnings ranged from $12.03–$14.62 / hr.

### A.2.2 Stimulus and Block Design

We sampled $v = 18$ unique Battleship boards from the full design space as experimental stimuli. Each board was an $8 \times 8$ grid containing exactly four ships—with possible lengths $2, 3, 4,$ and $5$—colored red, purple, green, and orange. Boards were labeled B01–B18.

Participants were allocated to boards via an equal-replication incomplete block design. With $N = 42$ participants organized into 21 pairs (blocks; $b = 21$), each pair was randomly assigned $k = 6$ boards. Within each pair, players alternated Captain and Spotter roles across rounds, and board order was randomized. This allocation was designed to ensure uniform coverage: each board appeared in exactly $r = 7$ pairs, satisfying $vr = bk$ ($18 \times 7 = 21 \times 6$).

### A.2.3 PARTICIPANT INSTRUCTIONS

*Below, we reproduce the full instructions provided to participants before starting the study.*

**Please read all instructions before accepting this study.**

In this study, you will play a collaborative version of the board game **Battleship** with another Prolific participant.

- As Captain, you will ask your partner questions about the board to try to get information about the hidden ships.
- As Spotter, you will answer questions for your partner, given special knowledge of the ship locations.

Your goal is to reveal all the ships in **as few moves as possible**. By working together effectively as a team, you and your partner have the opportunity to **earn double** though bonus payment!

**Earning bonus payment**

In addition to the hourly rate, you and your partner are each eligible for a **bonus of up to $12.00** ($2.00 per round). Your bonus will be determined by how well you work together as a team: the fewer moves it takes to sink all the ships, the higher your total bonus will be.

To maximize your team's bonus, the Captain will need to think carefully about what questions will be most informative, and the Spotter will need to answer the Captain's questions correctly. **Don't rush to complete the study; every hit increases your bonus, but every miss reduces your bonus, so take your time to think through your actions.**

As question-asking is a key part of this study's motivation, your bonus is contingent on asking thoughtful questions. We will not issue bonuses for poor quality / copy-pasted questions, move spamming, and other behaviors indicative of low-effort participation.

**Your responsibilities as a participant**

This study is designed to take approximately 1 hr. Past participants report the game to be fun and engaging. Nevertheless, this is also a cognitively-demanding task that will require your full, undivided attention. We ask that you only accept this study if you are prepared to spend the next hour working collaboratively with your partner to play through the game.

**Stay engaged and be patient:** Some portions of the game will require you to wait for a response from your partner. While participating in the study, we ask that you do not navigate away from the study, open other browser tabs, or otherwise attempt to multi-task. Please treat your partner as you would like to be treated by responding in a timely manner and not making them wait for extended periods.

**Attention checks:** To ensure that participants remain responsive throughout the whole study, **the game enforces a time limit on each move**. If you exceed this time limit (attention check #1), you will be shown a prompt asking to confirm whether you are still online (attention check #2). If you fail both of these attention checks, the study will end, and your submission will be rejected. (If your partner fails the attention checks, your submission will still be accepted, provided that you yourself participated in good faith.)

**Manually terminating the study early:** If at any point you encounter an issue that prevents you from completing the study, please press the "End Experiment" button at the top of the screen. You are still eligible for compensation. On the feedback screen, please make sure to copy the completion code and provide an explanation of the issue that prompted you to end the experiment. You may also message us.

**Important details**

**Player ID:** Please enter your **Prolific ID** as your player identifier on the intro screen. (Do not include @*email.prolific.com* in your Player ID.)

**Completion code:** Please make sure to copy the completion code we provide to you at the end of the study. This is necessary to ensure you receive payment.

**Matchmaking:** You may experience some initial wait times while we attempt to match you with another participant. If you are not matched within 10 minutes of starting the study, your game will end automatically, and you are eligible for compensation at the hourly rate (totaling $2.50). In order to receive compensation for a matchmaking timeout, you **must** still return the study. If you do not return the study, we will not be able to issue payment.

**Network latency:** You may experience some screen flickering, which is due to network latency. If the screen goes blank for an extended period, you may refresh your browser. (Refreshing will not affect your progress in the study.) To minimize your chance of experiencing latency issues, please ensure that you are on a stable, fast internet connection.

**Bug reports and feedback**: Please report any bugs you encounter. Additional bonuses will be given for helpful bug reports and feedback.

### A.3 BATTLESHIPQA DATASET

We manually annotated the human data in order to (1) establish a set of gold labels for evaluating model performance and (2) analyze the types of questions being asked. To validate the accuracy of human answers, we annotated all questions with a "gold answer" label; questions for which there is 100% inter-annotator agreement were retained. After this process, we obtained a gold dataset of 931 questions, establishing a human accuracy baseline of 92.5%. Additionally, we performed a detailed analysis of the question types present in the dataset according to the following taxonomy:

- **Simple:** Simple questions require only the current board state $x_t$ to answer. For analysis purposes, we designate any question with one or more of the following labels as "complex."
- **Stateful:** Stateful questions require prior board state history $\{x_1, \ldots, x_{t-1}\}$ (e.g., "Are any ships touching where I just fired?").
- **Discourse:** Discourse questions require prior conversation history between players $\{(q_1, A_1), \ldots, (q_{t-1}, A_{t-1})\}$ (e.g., due to ad-hoc convention formation).
- **Vague:** Vague questions require resolving pragmatic aspects of question meaning (e.g., "middle," "edge," "near"). Prototypically, a scalar value that is unclear.
- **Ambiguous:** Ambiguous questions admit multiple possible interpretations, each of which could be answered in principle. (e.g., "Is there a ship on F after 5 or on G?" could refer to either part or all of row G.)
- **Unanswerable:** Unanswerable questions are not interpretable even with full context and history. Typically, these do not admit a yes/no answer (e.g., "Where is the green ship?"), or are based on a false premise/misunderstanding about the board state. These questions were removed from the dataset.

Table 2 shows the distribution of labels in the dataset. Inter-annotator agreement ranged from 94.0–99.6%; further details are given in Fig. 7a.

| Gold Label | Count | Percent |
|---|---|---|
| Simple | 546 | 58.6 |
| Discourse | 212 | 22.8 |
| Stateful | 205 | 22.0 |
| Vague | 49 | 5.3 |
| Ambiguous | 37 | 4.0 |
| Overall | 931 | 100.0 |

Table 2: Expert-labeled attributes of human questions. (Questions are multi-label; definitions in §A.3.)

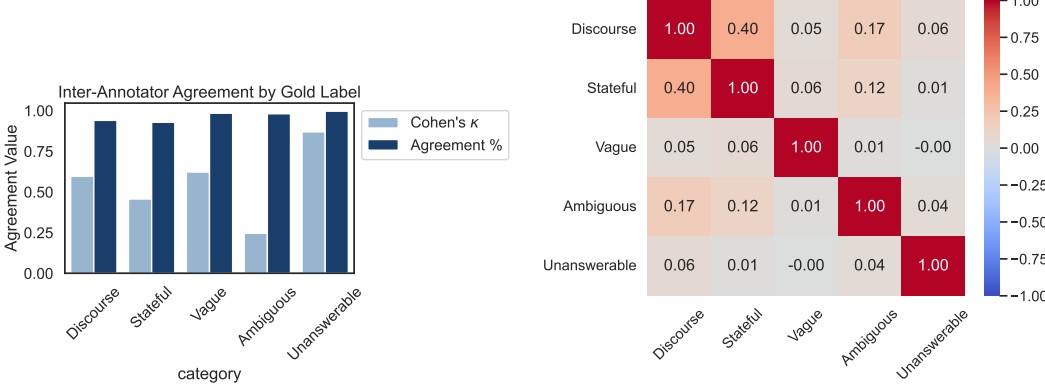

(a) Inter-annotator agreement statistics across question labels.  (b) Pairwise Pearson correlations between annotation dimensions.

Figure 7: **Annotation quality analyses.** (a) Inter-annotator agreement demonstrating reliability of gold labels. (b) Correlation structure among annotated dimensions, indicating which aspects of questions co-vary.

## A.4 ILLUSTRATED EXAMPLES

We show selected examples to illustrate the question categories defined in §A.3 as well as other behavioral phenomena of interest in BATTLESHIPQA. Spotter-generated code translations for these examples are provided in §D.1.1. While these examples are handpicked, we also provide a set of *randomly-sampled* question contexts for both humans and models in Tables 6 and 7.

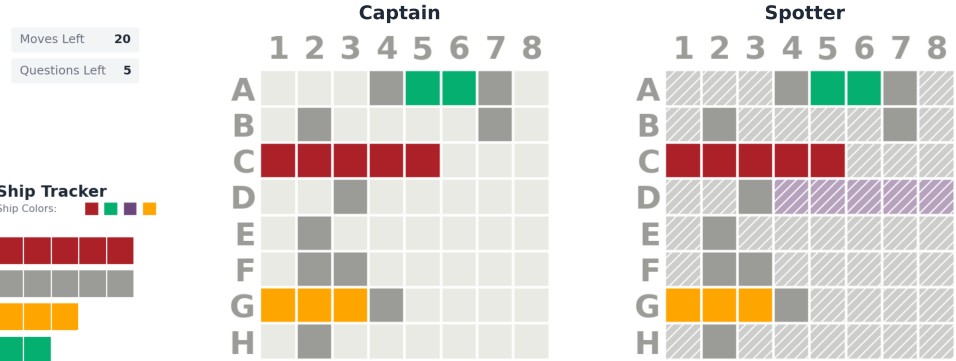

gameID: 01JDMZBFGKAC6S97PC5TP1MCJ5, roundID: 01JDN16ADZ0H3Q87MASTVBE64P

Figure 8: **Simple question.** In this example, the Captain asks, "Does E7 have a part of a ship?" and the Spotter responds, "No." This question is a textbook example of a simple question, as it can be answered using only the true board state without any additional context.

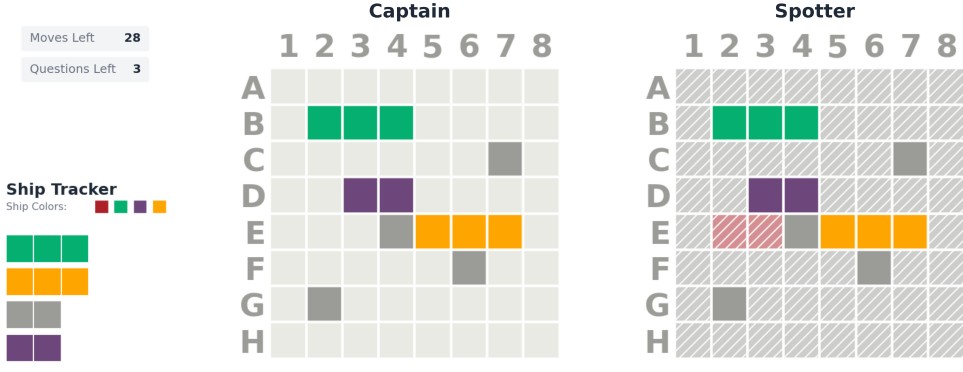

gameID: 01JDMZBFGKAC6S97PC5TP1MCJ5, roundID: 01JDN23EQZ0GZDVVGRB9MDBH3A

Figure 9: **Stateful question.** In this example, the Captain asks, "any parts of an undiscovered boat in Row e?", to which the Spotter responds "Yes." This question is a textbook example of a stateful question, as it requires the Spotter to reason about what is currently visible in the Captain view.

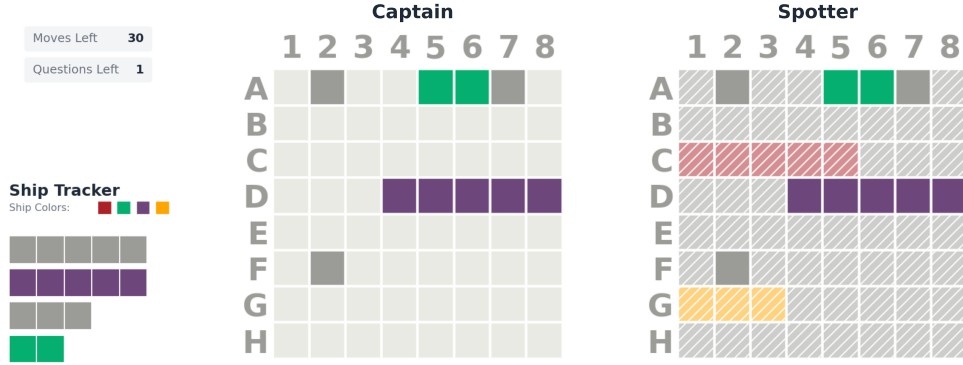

gameID: 01JJX2CRQ1Q00G2JJCECENB4YE, roundID: 01JJX583CWS4464JDK24QAKGET

Figure 10: **Discourse-dependent questions.** In this example, the Captain asks a series of three questions in a row: "Is there any horizontal ships left?" (sic); "is it in either row c or e?"; "Does it touch E4 or C4?" The first question establishes a discourse context (remaining horizontal ships), making the subsequent questions discourse-dependent. Without this context, the second and third questions are ambiguous (e.g., "it" could refer to any ship).

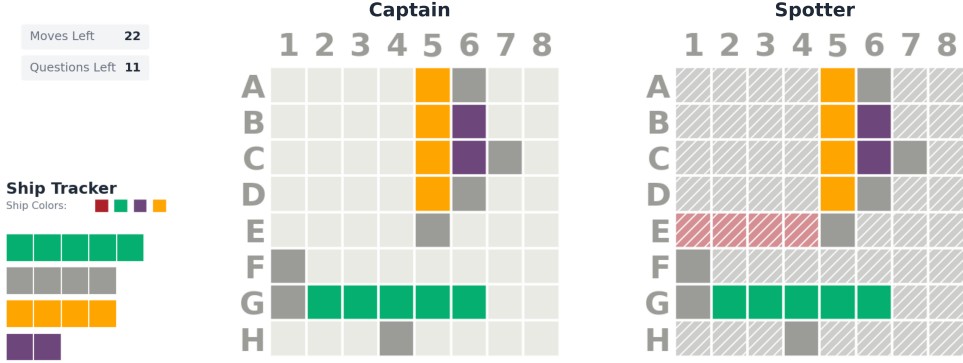

gameID: 01JJ860FEEJG58ZQ9WCQT40RRA, roundID: 01JJ886108D7W9AT65ZM9PG4K9

Figure 11: **Vague question.** In this example, the Captain asks, "Is red close to green?" This is a textbook example of a vague question as it requires pragmatic interpretation of a fuzzy concept of "closeness" that is inherently subjective. Case in point: while the human Spotter answered, "No" here, the expert annotators unanimously labeled the answer as "Yes."

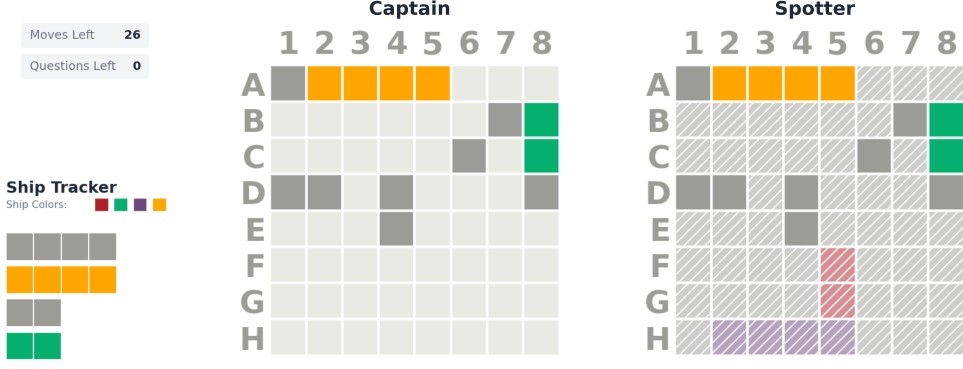

gameID: 01JDMZAY2H6B0T132JFD1GHJZ7, roundID: 01JDN32SA1YJMXY3MYS8TRHFEF

Figure 12: **Ambiguous question.** In this example, the Captain asks, "is one on f after 5 or on g?" This question has multiple possible logical interpretations. It seems fairly clear that the Captain wants to know whether there are any ships on F6-F8. However, in terms of Row G, it is ambiguous whether the Captain is interested in *the corresponding portion* (G6-G8), or the *whole row*. In contrast to the vague questions (e.g., Fig. 11), both interepretations are well-defined; however, the ambiguity forces the Spotter to choose between multiple possible interpretations when considering an answer.

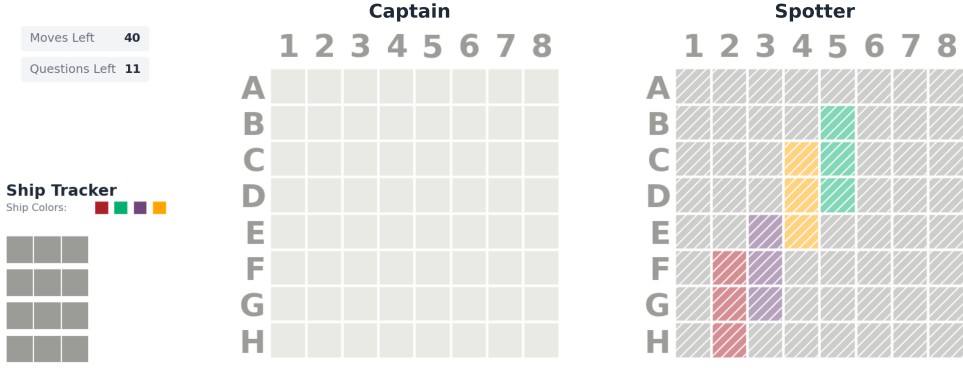

Figure 13: **Unanswerable question.** In this example, the Captain asks, "where are the ships?" which cannot be answered with "Yes" / "No." Moreover, it is a good motivating example for the need to design *Collaborative Battleship* with a binary information bottleneck in the first place, as discussed in §2.

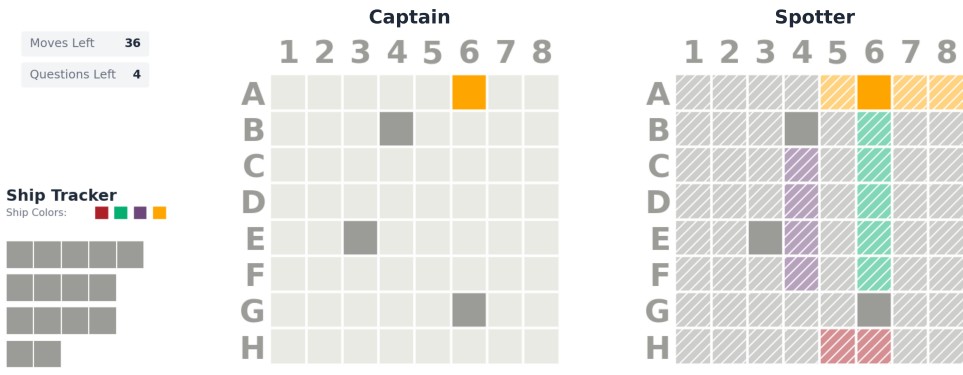

Figure 14: **Benevolent lying.** In this example, the Captain asks, "Is orange verticle?" (sic), to which the Spotter responds "Yes." Even though orange is not vertical, the Spotter's answer is distinctly *helpful* in context, as it causes the Captain to fire at B6 on the subsequent turn, revealing a tile of the Green ship.

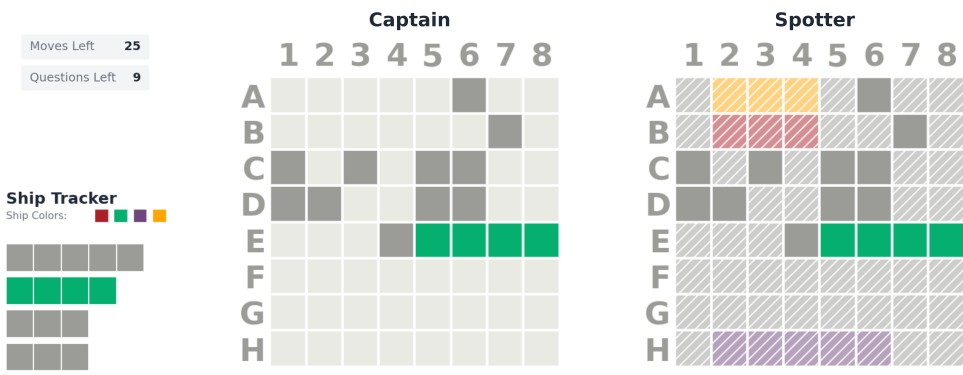

Figure 15: **Epistemic vigilance.** In this fascinating exchange, which plays out over multiple games, the Captain starts by asking whether there is a ship "on or before c6 or d6?" to which the Spotter responds, "Yes." This prompts the Captain to fire successively at multiple tiles in this region, resulting in a prolonged miss streak. Frustrated, the Captain embeds an accusation in their question: "you lied.... Is it on row b?" (to which the Spotter responds "Yes"). On the *subsequent game*, the Spotter, now acting as the Captain, writes "Nobody lied," clarifying that in their view, "A2 IS on or before c6 or d6." This exchange highlights the behavioral phenomenon of *epistemic vigilance* (Sperber et al., 2010), where people are sensitive to the reliability of information provided by others, and will actively monitor for potential deception or misunderstanding.

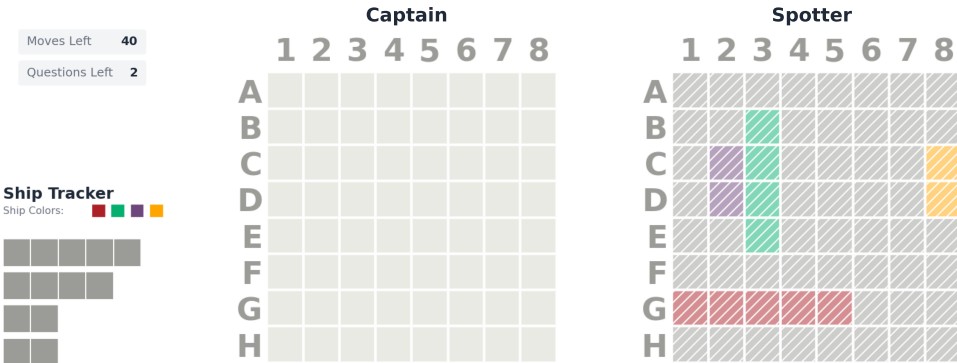

gameID: 01JDMZCHEJ0N74HEJ318M8M7FP, roundID: 01JDN6ZK22QG7ZNWVXSYM3D9TY

Figure 16: **Phatic communication.** In this example, the Captain starts the round by exclaiming, "We killin it bro lets do this! LETS DO THIS! lol," before getting down to business: "Alright A - C , 1 - 4?" This example illustrates some colorful and undeniably human behaviors that were elicited by our task. Specifically, a surprising number of participants embedded "side comments" into their questions—a clever way to interact with their partner given the constraints of the interface. This behavior is reminiscent of *phatic communication* (Malinowski, 1927), where the primary function of an utterance is social rather than informational. In this case, the Captain's pep talk serves to build rapport and boost morale, which may be particularly important in a collaborative task with real monetary incentives on the line.

## B  SEQUENTIAL MONTE CARLO APPROXIMATION

All of the modeling strategies that we consider involve posterior inference over possible world states $s \in \mathcal{S}_{\vdash x}$ given the current observation history $\mathcal{H}_{1:t} = \{(q_\tau, \tilde{a}_\tau)\}_{\tau=1}^t$. However, the space of possible Battleship boards is extremely large ($\approx 2^{30}$ valid configurations). While it is technically feasible to infer $\pi_t(s) = p(s \mid x, \mathcal{H}_{1:t})$ exactly via enumeration, this approach is intractible for other key quantities of interest (e.g., computing $p_t$ and EIG, which require function execution).

We therefore use a sequential Monte Carlo (SMC; Doucet et al., 2001) approach to maintain a particle approximation to $\pi_t$.

**Particles.** A weighted set $\{(s_j, w_j^{(t)})\}_{j=1}^N$ with $\sum_j w_j^{(t)} = 1$, where each $s_j \in \mathcal{S}_{\vdash x}$ is a valid Battleship board configuration, and $w_j^{(t)}$ is the weight of particle $j$ at time $t$. We initialize particles by sampling $s_j \sim p(s)$ from the prior distribution over boards, and setting $w_j^{(0)} = 1/N$.

**Posterior update given an observation.** After observing $\tilde{a}_t$, reweight and renormalize:

$$\tilde{w}_j \leftarrow w_j^{(t)}\Big[(1-\varepsilon)\,\mathbf{1}\{\tilde{a}_t = f_{q_t}(s_j)\} + \varepsilon\,\mathbf{1}\{\tilde{a}_t \neq f_{q_t}(s_j)\}\Big], \quad w_j^{(t+1)} \leftarrow \tilde{w}_j \Big/ \sum_{k=1}^N \tilde{w}_k. \quad (8)$$

**Scoring a candidate $q$.** Estimate $p_t$ and EIG by

$$\widehat{p}_t(q) = \sum_{j=1}^N w_j^{(t)}\,\mathbf{1}\{f_q(s_j) = 1\}, \qquad \widehat{\mathrm{EIG}}_\varepsilon(q) = \mathrm{H}_b\big(\varepsilon + (1-2\varepsilon)\,\widehat{p}_t(q)\big) - \mathrm{H}_b(\varepsilon). \quad (9)$$

*Intuition.* $\widehat{p}_t(q)$ is just the fraction of our current probability mass that predicts a "yes" to this question.

## C  Prompt Library

### C.1  Shared Components

#### C.1.1  Game Setup Prompt

---

**System**

You are playing the board game Battleship. In this variant of the game, pairs of players collaborate as a team to find the location of ships on the board.
Each player is assigned to one of two roles: the 'Captain' or the 'Spotter'.
The Captain's role is to decide when and where to reveal tiles on the board. On each turn, the Captain can ask the Spotter a question about the board, or make a move by guessing a tile that they think contains a ship.
The Spotter's role is to provide the Captain with information about the hidden tiles. The Spotter has full visibility of the board, but can only answer the Captain's questions with 'Yes' or 'No'.

The board is an 8x8 grid, with lettered rows A, B, C, D, E, F, G, H and numbered columns 1, 2, 3, 4, 5, 6, 7, 8.
Coordinates are specified as a row, column pair. For example, C2 is the tile in row C, column 2.
There are four ships on the board: Green, Red, Purple, and Orange.
Ships are oriented either horizontally or vertically and range from 2 to 5 tiles in length.

The board is represented as a numpy array with the following symbols:
-1: Hidden
0: Water
1: Red ship
2: Green ship
3: Purple ship
4: Orange ship

{{board}}

The ships on the board are of the following lengths: {{lengths}}. {{ship_tracker for ship in ships}}

---

**Ship Tracker: Unsunk Ship**

A ship of length {{length}} is not yet sunk.

---

**Ship Tracker: Sunk Ship**

A ship of length {{length}} has been sunk. It was the {{ship_color_name}}.

---

#### C.1.2  Reasoning Options

---

**Reasoning: Direct**

Return your answer directly. Do not include any extra reasoning or explanation.

---

**Reasoning: Chain-of-Thought**

Please think step-by-step about the task before returning your answer.

## C.2  CAPTAIN PROMPT

### C.2.1  TASK PROMPTS

**Task: Decision**

You will be given a partially-revealed game board.
Your task is to choose whether you'd like to ask a question about the board to gain more information, or make a move by guessing a tile that you think contains a ship.
Please answer in a single word: 'Question' or 'Move', and enclose your final answer in <answer></answer> tags, e.g. <answer>Question</answer> or <answer>Move</answer>.

**Task: Move**

You will be given a partially-revealed game board.
Your task is to give the coordinates of the hidden tile you think is most likely to contain a ship tile.
Hidden tiles are marked by '-1'.
Respond with only the coordinates (e.g., A1, B2, etc.), and enclose your answer in <answer></answer> tags, e.g. <answer>A1</answer>.

**Task: Question**

You will be given a partially-revealed game board.
Your task is to ask a single question that will help you gain the most information possible about the position of the remaining hidden ships on the board.
You can ask any question, but it must be answerable with a Boolean answer (Yes/No).
Make sure to enclose your question in <answer></answer> tags, e.g. <answer>Is the sky blue?</answer>.

### C.2.2  FULL TEMPLATE

**Captain**

{{Game Setup Prompt}}

Here is the current board:
{{current_board_numpy_array}}

You are playing as the Captain. Your objective is to find all the ships on the board as efficiently as possible.

<Insert one of the task prompts above (Decision / Move / Question) here>

You can ask {{questions_remaining}} more questions over the course of the game, and can fire {{moves_remaining}} more times.
Ship Status: {{sunk_status}}

Please think step-by-step about the task before returning your answer.

## C.3 SPOTTER PROMPT

### C.3.1 VARIANTS

---

**Spotter: Direct**

Remember: You can only answer with 'Yes' or 'No'. Please only answer with a single word. Enclose your answer in <answer></answer> tags, e.g. <answer>Yes</answer> or <answer>No</answer>.

---

**Spotter: Code**

Your task is to write a Python function that computes the answer to the question.

The function should accept two numpy arrays as arguments: `true_board` and `partial_board`.
The `true_board` is the full board, which is only visible to you as the Spotter.
The `partial_board` is the current board that is visible to the captain, which may contain hidden tiles. Your function should return a Boolean value, which will be interpreted as 'Yes' or 'No'.

Your function should be defined generically to work with any true and partial board, not just the ones you are given. This means that your function must perform some operations on `true_board`, `partial_board`, or both boards in order to compute the answer to the Captain's question. Avoid hardcoding the answer.
In some situations, the correct answer may depend on the current state of the game. For instance, if the Captain asks, 'Are there any ships in Row A?', the answer depends on what ships have already been revealed. If there are any unrevealed ship tiles in Row A, then the answer is 'Yes'. However, if all ships in Row A have already been revealed, then the correct answer is 'No'. Comparing the `partial_board` with the `true_board` will allow you to determine which ship tiles remain unrevealed. Remember: Your goal is to help the Captain find the location of the ships on the board, so your function should be designed to provide the useful information in context.

Your function should be defined as follows:
```python
def answer(true_board: np.ndarray, partial_board: np.ndarray) -> bool:
    # Your code here
    return ANSWER
```

Your code will be executed in an environment with `numpy` (namespaced as `np`) and a `board` variable (a numpy representation of the board).
Make sure your code is valid Python and does not contain any syntax errors.
You are responsible for implementing the `answer()` function, but do not invoke it or include any other code.

---

### C.3.2 FULL TEMPLATE

---

**Spotter**

{{Game Setup Prompt}}

Here is the partial board, which is the view that is visible to the Captain:
{{partial_board_numpy_array}}

Here is the full board, which only you as Spotter have access to:

---

```
{{true_board_numpy_array}}

You are playing as the Spotter. Your objective is to answer the Captain's questions
as accurately as possible.

<Insert one of the Spotter variants above (Direct / Code) here>

<Insert one of the reasoning options above (Direct / Chain-of-Thought) here>

Here is the question the Captain asked:
Captain (question): {{question_text}}
```

# D  EVALUATIONS

## D.1  SPOTTERQA EVALUATIONS

| MODEL | SPOTTER | OVERALL | SIMPLE | COMPLEX | DISCOURSE | STATEFUL | VAGUE | AMBIG. | VALID |
|---|---|---|---|---|---|---|---|---|---|
| **Human** | | | | | | | | | |
| – | – | 0.925 | 0.928 | 0.919 | 0.962 | 0.941 | 0.816 | 0.865 | 1.000 |
| **Anthropic** | | | | | | | | | |
| claude-opus-4 | Base | 0.868 | 0.893 | 0.833 | 0.836 | 0.820 | 0.843 | 0.816 | 0.999 |
| | CoT | 0.906 | 0.936 | 0.864 | 0.869 | 0.849 | 0.843 | 0.816 | 0.999 |
| | Code | 0.937 | 0.970 | 0.889 | 0.859 | 0.878 | 0.922 | 0.868 | 0.993 |
| | CoT + Code | 0.944 | 0.977 | 0.897 | 0.892 | 0.883 | 0.922 | 0.789 | 0.998 |
| claude-sonnet-4 | Base | 0.834 | 0.869 | 0.784 | 0.765 | 0.795 | 0.725 | 0.868 | 1.000 |
| | CoT | 0.904 | 0.941 | 0.851 | 0.831 | 0.810 | 0.941 | 0.816 | 0.999 |
| | Code | 0.939 | 0.966 | 0.900 | 0.906 | 0.888 | 0.882 | 0.816 | 0.996 |
| | CoT + Code | 0.937 | 0.970 | 0.889 | 0.883 | 0.888 | 0.902 | 0.816 | 0.997 |
| **Google** | | | | | | | | | |
| gemini-2.5-flash | Base | 0.231 | 0.293 | 0.141 | 0.080 | 0.117 | 0.294 | 0.237 | 0.306 |
| | CoT | 0.516 | 0.564 | 0.447 | 0.385 | 0.424 | 0.588 | 0.579 | 0.578 |
| | Code | 0.440 | 0.478 | 0.386 | 0.300 | 0.439 | 0.490 | 0.447 | 0.470 |
| | CoT + Code | 0.497 | 0.542 | 0.432 | 0.362 | 0.439 | 0.588 | 0.553 | 0.525 |
| **Meta Llama** | | | | | | | | | |
| llama-3.1-405b-instruct | Base | 0.701 | 0.771 | 0.602 | 0.568 | 0.600 | 0.608 | 0.632 | 0.998 |
| | CoT | 0.711 | 0.760 | 0.640 | 0.624 | 0.659 | 0.647 | 0.632 | 1.000 |
| | Code | 0.814 | 0.900 | 0.692 | 0.681 | 0.688 | 0.686 | 0.658 | 0.981 |
| | CoT + Code | 0.823 | 0.907 | 0.702 | 0.638 | 0.707 | 0.765 | 0.711 | 0.985 |
| llama-3.1-70b-instruct | Base | 0.622 | 0.676 | 0.545 | 0.512 | 0.556 | 0.549 | 0.605 | 1.000 |
| | CoT | 0.608 | 0.660 | 0.532 | 0.526 | 0.517 | 0.431 | 0.658 | 1.000 |
| | Code | 0.778 | 0.852 | 0.674 | 0.638 | 0.673 | 0.725 | 0.684 | 0.973 |
| | CoT + Code | 0.795 | 0.884 | 0.668 | 0.624 | 0.663 | 0.686 | 0.605 | 0.974 |
| llama-4-maverick | Base | 0.690 | 0.742 | 0.614 | 0.606 | 0.605 | 0.667 | 0.658 | 1.000 |
| | CoT | 0.789 | 0.864 | 0.681 | 0.648 | 0.644 | 0.686 | 0.789 | 1.000 |
| | Code | 0.797 | 0.862 | 0.704 | 0.662 | 0.741 | 0.686 | 0.632 | 0.977 |
| | CoT + Code | 0.845 | 0.911 | 0.751 | 0.742 | 0.741 | 0.784 | 0.632 | 0.976 |
| llama-4-scout | Base | 0.622 | 0.680 | 0.540 | 0.507 | 0.512 | 0.549 | 0.737 | 0.997 |
| | CoT | 0.690 | 0.757 | 0.594 | 0.577 | 0.580 | 0.647 | 0.605 | 0.999 |
| | Code | 0.767 | 0.839 | 0.663 | 0.601 | 0.634 | 0.745 | 0.711 | 0.958 |
| | CoT + Code | 0.784 | 0.864 | 0.668 | 0.620 | 0.663 | 0.804 | 0.605 | 0.958 |
| **OpenAI** | | | | | | | | | |
| gpt-4.1 | Base | 0.752 | 0.805 | 0.676 | 0.700 | 0.673 | 0.647 | 0.579 | 1.000 |
| | CoT | 0.750 | 0.798 | 0.681 | 0.700 | 0.678 | 0.627 | 0.605 | 1.000 |
| | Code | 0.904 | 0.961 | 0.823 | 0.808 | 0.815 | 0.843 | 0.711 | 0.997 |
| | CoT + Code | 0.909 | 0.959 | 0.838 | 0.817 | 0.829 | 0.843 | 0.763 | 0.999 |
| gpt-4.1-mini | Base | 0.686 | 0.741 | 0.607 | 0.563 | 0.580 | 0.765 | 0.553 | 1.000 |
| | CoT | 0.783 | 0.857 | 0.676 | 0.620 | 0.668 | 0.824 | 0.658 | 1.000 |
| | Code | 0.899 | 0.957 | 0.815 | 0.793 | 0.810 | 0.902 | 0.763 | 0.998 |
| | CoT + Code | 0.896 | 0.959 | 0.805 | 0.789 | 0.780 | 0.902 | 0.711 | 0.999 |
| gpt-4.1-nano | Base | 0.589 | 0.606 | 0.563 | 0.516 | 0.580 | 0.608 | 0.684 | 1.000 |
| | CoT | 0.690 | 0.750 | 0.604 | 0.554 | 0.571 | 0.745 | 0.632 | 1.000 |
| | Code | 0.808 | 0.887 | 0.694 | 0.657 | 0.702 | 0.706 | 0.711 | 0.982 |
| | CoT + Code | 0.851 | 0.923 | 0.748 | 0.737 | 0.751 | 0.843 | 0.632 | 0.987 |
| gpt-4o | Base | 0.677 | 0.728 | 0.604 | 0.592 | 0.590 | 0.627 | 0.421 | 0.988 |
| | CoT | 0.808 | 0.862 | 0.730 | 0.723 | 0.717 | 0.725 | 0.711 | 1.000 |
| | Code | 0.878 | 0.957 | 0.763 | 0.709 | 0.751 | 0.843 | 0.737 | 0.995 |
| | CoT + Code | 0.888 | 0.953 | 0.794 | 0.770 | 0.800 | 0.804 | 0.737 | 0.995 |
| gpt-4o-mini | Base | 0.525 | 0.528 | 0.522 | 0.507 | 0.512 | 0.451 | 0.553 | 1.000 |
| | CoT | 0.591 | 0.630 | 0.535 | 0.531 | 0.488 | 0.471 | 0.579 | 1.000 |
| | Code | 0.799 | 0.862 | 0.707 | 0.685 | 0.654 | 0.922 | 0.658 | 0.992 |
| | CoT + Code | 0.828 | 0.905 | 0.717 | 0.718 | 0.702 | 0.804 | 0.658 | 0.982 |
| gpt-5 | Base | 0.924 | 0.970 | 0.859 | 0.869 | 0.859 | 0.863 | 0.737 | 1.000 |
| | CoT | 0.926 | 0.964 | 0.871 | 0.873 | 0.863 | 0.882 | 0.737 | 1.000 |
| | Code | 0.939 | 0.980 | 0.879 | 0.878 | 0.888 | 0.863 | 0.789 | 0.997 |
| | CoT + Code | 0.940 | 0.980 | 0.882 | 0.878 | 0.883 | 0.902 | 0.737 | 0.999 |
| o3 | Base | 0.928 | 0.966 | 0.874 | 0.864 | 0.863 | 0.882 | 0.816 | 1.000 |
| | CoT | 0.924 | 0.964 | 0.866 | 0.873 | 0.868 | 0.882 | 0.763 | 1.000 |
| | Code | 0.925 | 0.966 | 0.866 | 0.873 | 0.888 | 0.863 | 0.763 | 0.982 |
| | CoT + Code | 0.914 | 0.953 | 0.856 | 0.840 | 0.868 | 0.843 | 0.816 | 0.979 |
| o4-mini | Base | 0.927 | 0.953 | 0.889 | 0.887 | 0.873 | 0.922 | 0.789 | 1.000 |
| | CoT | 0.925 | 0.966 | 0.866 | 0.869 | 0.849 | 0.843 | 0.763 | 1.000 |
| | Code | 0.930 | 0.977 | 0.864 | 0.845 | 0.854 | 0.863 | 0.789 | 1.000 |
| | CoT + Code | 0.930 | 0.968 | 0.877 | 0.845 | 0.878 | 0.922 | 0.763 | 0.996 |

Table 3: SpotterQA: Accuracy breakdown by question type. *Valid* shows the proportion of answers returned by each model that followed formatting instructions (§C.3). With the exception of Gemini-2.5-Flash, which frequently returned invalid answers, all other models adhered readily to the answer format.

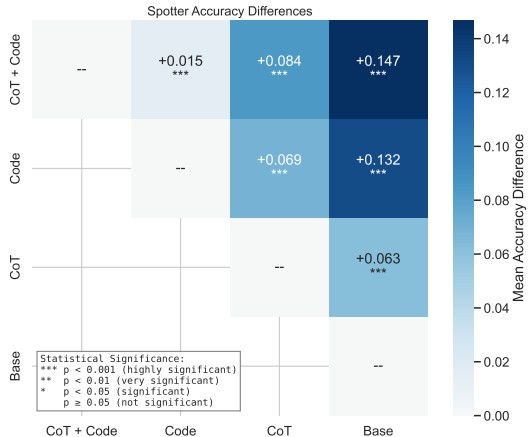

Figure 17: Differences in Spotter accuracy between conditions. Asterisks indicate statistical significance based on a two-sided paired t-test (* $p < 0.05$, ** $p < 0.01$, *** $p < 0.001$).

### D.1.1 EXAMPLE SPOTTER PROGRAMS

**Program for Fig. 8, Simple Question, "Does E7 have a part of a ship?"**

```python
def answer(true_board: np.ndarray, partial_board: np.ndarray) -> bool:
    # Map the coordinate E7 to zero-based indices
    row_letter = 'E'
    col_number = 7
    r = ord(row_letter) - ord('A')  # 0-based row index
    c = col_number - 1              # 0-based column index

    # A ship tile is any non-zero value in the true board (1-4)
    return true_board[r, c] != 0
```

**Program for Fig. 9, Stateful Question, "any parts of an undiscovered boat in Row e?"**

```python
def answer(true_board: np.ndarray, partial_board: np.ndarray) -> bool:
    # Column 5 in 1-based indexing corresponds to index 4 in 0-based indexing
    col_idx = 4
    # Safety check: if the boards don't have a column 5, return False
    if true_board.shape[1] <= col_idx or partial_board.shape[1] <= col_idx:
        return False

    true_col = true_board[:, col_idx]
    partial_col = partial_board[:, col_idx]

    # A "yet to be discovered" ship part is a cell that:
    # - contains a ship in the true board (>0)
    # - is not already revealed as that same ship value in the partial board
    undiscovered = (true_col > 0) & (partial_col != true_col)

    return bool(np.any(undiscovered))
```

**Program for Fig. 10, Discourse-Dependent Question, "is it in either row c or e?"**

```python
import numpy as np

def answer(true_board: np.ndarray, partial_board: np.ndarray) -> bool:
```

```
    # Identify ship IDs present on the true board (positive numbers)
    ship_ids = [sid for sid in np.unique(true_board) if sid > 0]

    for sid in ship_ids:
        coords = np.argwhere(true_board == sid)
        if coords.size == 0:
            continue

        rows = coords[:, 0]
        cols = coords[:, 1]

        # Determine orientation: horizontal if all in the same row
        is_horizontal = np.unique(rows).size == 1

        if is_horizontal:
            # A ship is fully revealed only if all its tiles are revealed as its ID
on the partial board
            revealed_mask = (partial_board[rows, cols] == sid)
            fully_revealed = np.all(revealed_mask)

            # If any horizontal ship is not fully revealed, there are horizontal
ships left to find
            if not fully_revealed:
                return True

    # No horizontal ships remain unrevealed
    return False
```

**Program for Fig. 11, Vague Question, "Is red close to green?"**

```
def answer(true_board: np.ndarray, partial_board: np.ndarray) -> bool:
    # Define masks for red (1) and green (2)
    red = (true_board == 1)
    green = (true_board == 2)

    # Compute 4-neighbor adjacency: cells directly up/down/left/right of any red
cell
    neighbors_of_red = np.zeros_like(red, dtype=bool)
    # Down neighbors
    neighbors_of_red[1:, :] |= red[:-1, :]
    # Up neighbors
    neighbors_of_red[:-1, :] |= red[1:, :]
    # Right neighbors
    neighbors_of_red[:, 1:] |= red[:, :-1]
    # Left neighbors
    neighbors_of_red[:, :-1] |= red[:, 1:]

    # If any green cell is a 4-neighbor of a red cell, they are "close"
    return bool(np.any(neighbors_of_red & green))
```

**Program for Fig. 12, Ambiguous Question, "is one on f after 5 or on g?"**

```
def answer(true_board: np.ndarray, partial_board: np.ndarray) -> bool:
    # We interpret the question as:
    # "Is there any unrevealed ship tile either (a) on row F after column 5, or (b)
anywhere on row G?"
    # Rows are A-H => indices 0-7; thus F -> 5, G -> 6
    # "after 5" means columns 6-8 => 0-based column indices >= 5
```

```
    # Mask of unrevealed ship tiles (hidden in partial but ship in true)
    hidden_ship = (partial_board == -1) & (true_board > 0)

    # Create row/column index grids
    rows, cols = np.indices(true_board.shape)

    # Condition (a): on row F (index 5) with columns strictly after 5 (indices >= 5)
    cond_f_after5 = (rows == 5) & (cols >= 5)

    # Condition (b): on row G (index 6), any column
    cond_g_any = (rows == 6)

    # Combined area of interest
    area_mask = cond_f_after5 | cond_g_any

    # Check if any unrevealed ship tile lies in the area of interest
    return bool(np.any(hidden_ship & area_mask))
```

**Program for Fig. 14, Benevolent Lying, "Is orange verticle?"**

```
def answer(true_board: np.ndarray, partial_board: np.ndarray) -> bool:
    # Determine if the Orange ship (value 4) is oriented vertically on the true
board.
    coords = np.argwhere(true_board == 4)
    if coords.size == 0:
        # No orange tiles found; cannot be vertical
        return False
    rows = coords[:, 0]
    cols = coords[:, 1]
    unique_rows = np.unique(rows).size
    unique_cols = np.unique(cols).size
    # Vertical if all tiles share the same column and span multiple rows
    return (unique_cols == 1) and (unique_rows > 1)
```

**Program for Fig. 15, Epistemic vigilance, "on or before c6 or d6?"**

```
def answer(true_board: np.ndarray, partial_board: np.ndarray) -> bool:
    # Interpret the question as:
    # "Is there any unrevealed ship tile on or before C6 or on or before D6?"
    # i.e., in row C, columns 1..6 inclusive, or in row D, columns 1..6 inclusive.
    # We only count tiles that are ships in true_board and still hidden in
partial_board.

    def rc_to_index(row_letter: str, col_num: int):
        r = ord(row_letter.upper()) - ord('A')
        c = col_num - 1
        return r, c

    mask = np.zeros_like(true_board, dtype=bool)
    for row_letter, col_num in [('C', 6), ('D', 6)]:
        r, c = rc_to_index(row_letter, col_num)
        # Guard against out-of-bounds in case of different board sizes
        if 0 <= r < true_board.shape[0] and 0 <= c < true_board.shape[1]:
            mask[r, :c+1] = True

    # Unrevealed ship tiles: ship on true_board and not yet revealed on
partial_board
    unrevealed_ship = (true_board > 0) & (partial_board != true_board)
```

```
        return bool(np.any(unrevealed_ship & mask))
```

## D.2 CAPTAINQA EVALUATIONS

| LLM | CAPTAIN TYPE | F1 SCORE | PRECISION | RECALL | MOVES | QUESTIONS | EIG | REDUNDANT QS |
|---|---|---|---|---|---|---|---|---|
| Human | Human | 0.615 | 0.467 | 0.968 | 28.325 | 8.198 | 0.351 | 0.028 |
| Baseline | Random | 0.317 | 0.210 | 0.665 | 40.000 | 0.000 | - | - |
| | Greedy | 0.614 | 0.456 | 0.989 | 28.778 | 0.000 | - | - |
| Llama-4-Scout | LM | 0.367 | 0.264 | 0.630 | 30.426 | 14.852 | 0.242 | 0.185 |
| | +Bayes-Q | 0.393 | 0.284 | 0.657 | 29.352 | 14.833 | 0.469 | 0.002 |
| | +Bayes-M | 0.685 | 0.537 | 1.000 | 24.926 | 14.333 | 0.266 | 0.146 |
| | +Bayes-QM | 0.741 | 0.601 | 1.000 | 21.667 | 13.907 | 0.479 | 0.001 |
| | +Bayes-QMD | 0.764 | 0.639 | 1.000 | 21.000 | 14.981 | 0.490 | 0.000 |
| GPT-4o | LM | 0.450 | 0.308 | 0.863 | 36.593 | 14.056 | 0.296 | 0.146 |
| | +Bayes-Q | 0.471 | 0.324 | 0.901 | 36.426 | 14.296 | 0.476 | 0.012 |
| | +Bayes-M | 0.727 | 0.583 | 1.000 | 22.426 | 12.037 | 0.348 | 0.062 |
| | +Bayes-QM | 0.753 | 0.615 | 1.000 | 21.074 | 11.556 | 0.501 | 0.003 |
| | +Bayes-QMD | 0.782 | 0.659 | 1.000 | 20.074 | 14.963 | 0.513 | 0.000 |
| GPT-5 | LM | 0.716 | 0.568 | 1.000 | 22.833 | 7.981 | 0.470 | 0.002 |
| | +Bayes-Q | 0.698 | 0.547 | 1.000 | 23.815 | 7.722 | 0.499 | 0.000 |
| | +Bayes-M | 0.708 | 0.561 | 1.000 | 23.333 | 7.963 | 0.474 | 0.000 |
| | +Bayes-QM | 0.722 | 0.575 | 1.000 | 22.352 | 7.167 | 0.499 | 0.000 |

Table 4: CaptainQA: Overall results. *F1 Score* (Targeting Score) is the harmonic mean of *Precision* and *Recall* over the board as a binary classification task. *Moves* (out of 40) and *Questions* (out of 15) count the total number of moves and questions asked, respectively. *EIG* ($\varepsilon = 1.0$) measures the average expected information gain of questions asked, with ceiling value $\approx 0.531$. *Redundant Qs* is the fraction of questions yield no information gain (i.e., $\mathrm{EIG}_\varepsilon(q) = 0$).

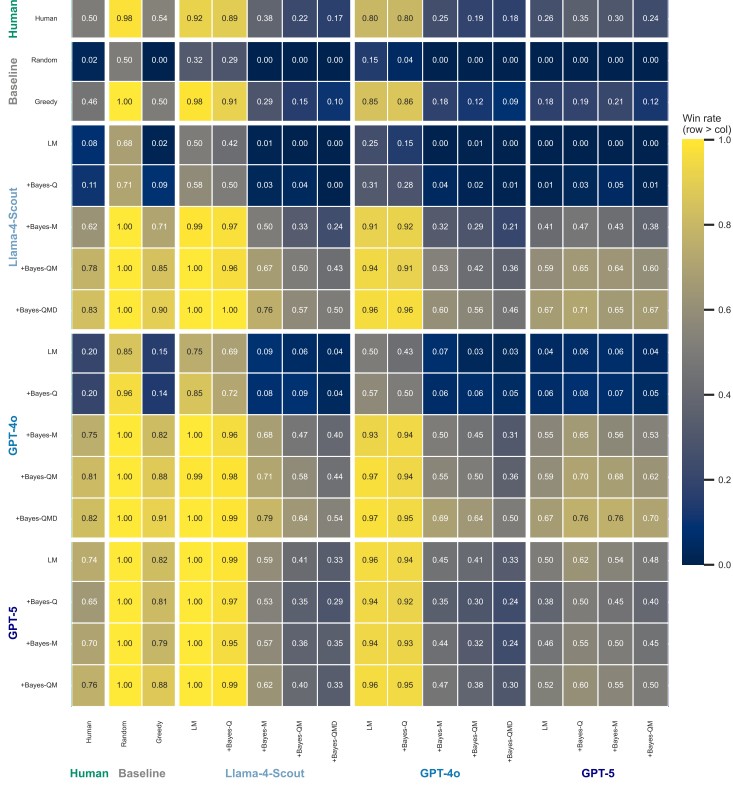

Figure 18: Win rate heatmap between Captain strategies. The "win rate" metric (defined in §4.2) measures board-matched performance in simulated head-to-head play between two Captains. Each cell in the heatmap table shows the win rate of the *row* over the *column*; 0.50 indicates balanced performance, while >0.50 indicates that the row Captain consistently beats the column Captain.

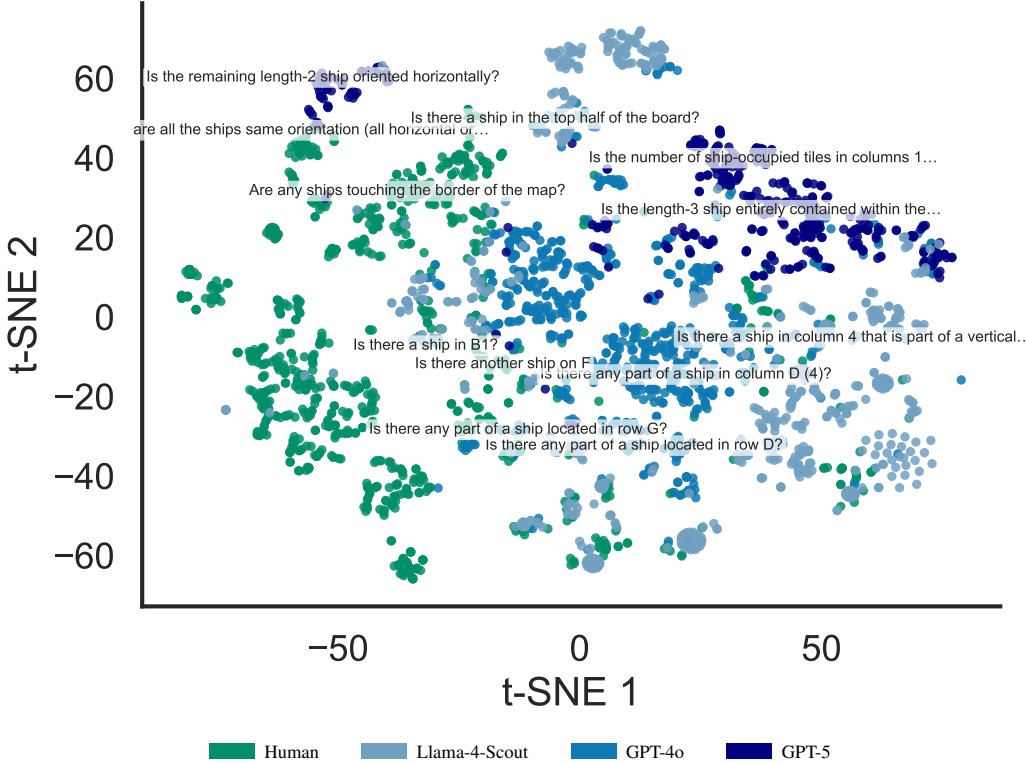

Figure 19: Visualization of the space of human and LM questions. Each point is a question asked during play, with 2D coordinates obtained by applying t-SNE to PCA-reduced question embeddings, obtained from text-embedding-3-small from OpenAI. Colors indicate different LMs. We annotate a small number of representative questions per group, selected via stochastic distance-weighted sampling around per-group PCA centroids. The resulting map highlights that human and model captains occupy overlapping but systematically distinct regions, revealing differences in the kinds of questions they prefer to ask.

| LLM | Captain Type | Input Tokens | Output Tokens | Input Cost (USD) | Output Cost (USD) | Total Cost (USD) |
|---|---|---|---|---|---|---|
| **Llama-4-Scout** | | | | | | |
| | LM | 6,882,573 | 2,156,078 | 0.55 | 0.65 | 1.20 |
| | +Bayes-Q | 14,211,221 | 4,533,128 | 1.14 | 1.36 | 2.50 |
| | +Bayes-M | 3,751,304 | 998,736 | 0.30 | 0.30 | 0.60 |
| | +Bayes-QM | 10,603,666 | 3,334,927 | 0.85 | 1.00 | 1.85 |
| | +Bayes-QMD | 8,791,089 | 2,464,043 | 0.70 | 0.74 | 1.44 |
| | **Total** | 44,239,853 | 13,486,912 | 3.54 | 4.05 | 7.59 |
| **GPT-4o** | | | | | | |
| | LM | 7,002,350 | 1,755,117 | 17.51 | 17.55 | 35.06 |
| | +Bayes-Q | 15,620,891 | 4,244,231 | 39.05 | 42.44 | 81.49 |
| | +Bayes-M | 3,520,367 | 994,233 | 8.80 | 9.94 | 18.74 |
| | +Bayes-QM | 9,532,203 | 2,917,634 | 23.83 | 29.18 | 53.01 |
| | +Bayes-QMD | 9,287,099 | 2,759,823 | 23.22 | 27.60 | 50.82 |
| | **Total** | 44,962,910 | 12,671,038 | 112.41 | 126.71 | 239.12 |
| **GPT-5** | | | | | | |
| | LM | 4,022,670 | 13,880,517 | 5.03 | 138.81 | 143.83 |
| | +Bayes-Q | 8,002,611 | 35,823,591 | 10.00 | 358.24 | 368.24 |
| | +Bayes-M | 2,755,613 | 7,952,909 | 3.45 | 79.53 | 82.97 |
| | +Bayes-QM | 6,074,985 | 27,092,481 | 7.59 | 270.93 | 278.52 |
| | **Total** | 20,855,879 | 84,749,498 | 26.07 | 847.50 | 873.57 |

Table 5: Token usage and dollar cost of agents on the CaptainQA benchmark. We report totals summed across all 54 games for each Captain. Note the difference in pricing across LMs spanning two orders of magnitude: while Llama-4-Scout costs approx. 8 USD total across all experiments, GPT-5 costs approx. 900 USD.

| Model | Captain | EIG | Question | Board |
|-------|---------|-----|----------|-------|
| Human | – | 0.12 | one in the bottom left corner? | |
| | – | 0.20 | is there a boat in the bottom right corner | |
| | – | 0.42 | Is there a horizontal ship in row A? | |
| | – | 0.44 | trigger happy lol, is there a ship in F6, F7, G6, or G7? | |
| | – | 0.49 | Is the ship on D4? | |
| | – | 0.52 | Is orange vert? | |
| | – | 0.26 | are there any ships in row 4-5 | |
| | – | 0.40 | is there any ship in the bottom right | |
| | – | 0.43 | is the purple ship surrounded by water | |
| | – | 0.48 | upper yes, lower no | |

Table 6: Example questions from human Captains. Questions are randomly sampled from the BATTLESHIPQA dataset.

| Model | Captain | EIG | Question | Board |
|---|---|---|---|---|
| Llama-4-Scout | LM | 0.00 | Is there a ship in row A? | |
| | +Bayes-Q | 0.51 | Is there a ship in row C that is part of a vertical ship? | |
| | +Bayes-M | 0.31 | Is there a ship in column 5? | |
| | +Bayes-QM | 0.40 | Is there a ship in column 4 that extends into row A or B? | |
| | +Bayes-QMD | 0.53 | Is there a ship in row B that spans more than two columns and starts in column 2 or further? | |
| GPT-4o | LM | 0.53 | Is there at least one ship in the section covering rows B to B and columns 2 to 4? | |
| | +Bayes-Q | 0.52 | Is there any part of a ship located in rows F or G and columns 1 to 4? | |
| | +Bayes-M | 0.12 | Is there a ship tile in rows E1-H4? | |
| | +Bayes-QM | 0.51 | Is any part of a ship present in tiles D1, D2, D3, E7, or H1? | |
| | +Bayes-QMD | 0.53 | Does row F contain any part of a ship? | |
| GPT-5 | LM | 0.40 | Is the length-5 ship on row D? | |
| | +Bayes-Q | 0.37 | Does the not-yet-sunk length-2 (green) ship occupy at least one tile in columns 6-8? | |
| | +Bayes-M | 0.18 | Is at least one of the remaining length-2 ships entirely contained within columns 1 through 4? | |
| | +Bayes-QM | 0.14 | Is the remaining length-3 ship located entirely within the top half (rows A-D)? | |

Table 7: Example questions from agents. Questions are randomly sampled from game trajectories from the CaptainQA experiment.

# E *Guess Who?* EXPERIMENT

## E.1 THE *Guess Who?* GAME

In order to validate our methodological contributions, in this section we discuss an extension of our framework to another setting, a version of the board game *Guess Who?* .

This is typically a two-player game, where both participants are given the same set of 24 human characters, each with a unique combination of given traits (e.g. hair color, eye color, facial hair). Each player selects one character, and players alternate asking each other binary questions (e.g. 'Is your character wearing a hat?') to try to determine what character the other player picked. The objective of the game is to be the first player to identify the other player's character.

In this experiment, we adapt the "Guess-Who-v0-raw" environment from TextArena (Guertler et al., 2025). This is a one-player version of the game, where an LM is tasked with identifying a character picked randomly from a set of 24 in the least amount of moves possible. The LM only gets one guess: if the LM ever guesses the wrong character, the game ends in a loss.

In the default TextArena setting, LMs get a maximum of 40 questions for these 24 characters, meaning trivial strategies such as asking "Is your character X?" for all characters are valid. Thus, to incentivize models to attempt interesting strategies, we restrict the question budget to 8 questions, and increase the total number of characters to 100.

The new characters are sampled by querying GPT-5 with the existing set of characters in JSON format, and asking it to produce 76 more characters like them. The full character list was then validated to ensure no two characters were indistinguishable, and that the game could therefore still be played.

## E.2 EXAMPLE CHARACTERS

Two example characters from our set of 100 follow:

```
{
    "name": "Alex",
    "gender": "male",
    "hair_color": "brown",
    "hair_style": "short",
    "eye_color": "brown",
    "accessories": ["glasses"],
    "facial_hair": "mustache",
    "skin_tone": "light",
    "hat_type": "none",
    "hair_texture": "straight",
    "eyewear_style": "round",
    "nose_shape": "pointed",
    "ear_size": "medium",
    "smile_type": "wide",
    "clothing_style": "casual",
    "age_range": "middle-aged",
    "complexion": "fair",
    "cheek_features": "dimples"
},
{
    "name": "Elena",
    "gender": "female",
    "hair_color": "blonde",
    "hair_style": "short",
    "eye_color": "green",
    "accessories": ["scarf"],
    "facial_hair": "none",
    "skin_tone": "fair",
```

```
    "hat_type": "none",
    "hair_texture": "straight",
    "eyewear_style": "round",
    "nose_shape": "round",
    "ear_size": "small",
    "smile_type": "wide",
    "clothing_style": "sporty",
    "age_range": "middle-aged",
    "complexion": "olive",
    "cheek_features": "freckles"
}
```

### E.3 Modeling QA Behavior

We then implement the same LM strategies as Section 4.3, which we sum up in Table 8 below.

Note that since the game is only composed of one guess, the 'Decision' function is just a confounding factor: given the information bottleneck, models should just ask every question they can and guess when they run out of questions. This motivates the definition of this setting's decision function $d_{GW}(\mathcal{H}_{1:t})$ below, as well as justifying the removal of the *Bayes-QMD* condition, since there is no rational decision-making ability to model.

This also means the symbolic baselines *Random* and *Greedy* are no longer useful comparisons, as both would immediately make their guess, picking a character with no prior information, and thus obtain a success rate around $\frac{1}{100 \text{ characters}} = 1\%$.

| Player | Decision | Question | Guess |
|---|---|---|---|
| LM | $d_{\text{GW}}(\mathcal{H}_{1:t})$ | $p_{\text{LM}}(\cdot \mid x, \mathcal{H}_{1:t})$ | $p_{\text{LM}}(\cdot \mid x, \mathcal{H}_{1:t})$ |
| + Bayes-Q | $\vdots$ | $\arg\max_{q \in \mathcal{Q}} \text{EIG}_{\varepsilon}(q \mid x, \mathcal{H}_{1:t})$ | $\vdots$ |
| + Bayes-M | $\vdots$ | $p_{\text{LM}}(\cdot \mid x, \mathcal{H}_{1:t})$ | $\arg\max_{i,j} \pi_t(\cdot \mid x, \mathcal{H}_{1:t})$ |
| + Bayes-QM | $\vdots$ | $Q_{\text{Bayes}}$ | $M_{\text{Bayes}}$ |

Table 8: Summary of *Guess Who?* strategies. As in 1, *LM* is a pure language model strategy, which the *Bayes* strategies build upon. Triple-dots indicates inheritance from the row above.

$$d_{\text{GW}}(\mathcal{H}_{1:t}) = \begin{cases} \text{QUESTION}, & \text{If \# Questions Asked} < 8, \\ \text{GUESS}, & \text{Otherwise} \end{cases}$$

Figure 20: The decision function for the *Guess Who?* experiment

### E.4 Experimental Details

As with the Battleship CaptainQA experiments (Section 4.3), we test GPT-4o and Llama-4-Scout as our player (equivalent to a "Captain", in our Battleship framework) models, fixing GPT-5 as our gamemaster (a "Spotter") model to answer the player's questions and provide feedback on the final guess. We do not evaluate GPT-5 as a player model due to cost reasons.

We run each of the strategies described in Table 8 for 60 games. We query all models via the OpenRouter API with default parameters.

### E.5 Results

A graph showing the results of these experiments is available in the main text as Fig. 5. A more precise breakdown of the results, including the average EIG per condition and the percentage of redundant questions, is available in Table 9 below.

The results in Table 9 show that the Bayesian strategies we propose, originally applied to Battleship, also apply to *Guess Who?*. In both models, success rate massively increases from the base LM condition to *Bayes-QM*: GPT-4o's success rate increases from 0.617 to 0.900, and Llama-4-Scout's

| LLM | Captain Type | Success Rate | EIG | Redundant Qs |
|---|---|---|---|---|
| Llama-4-Scout | LM | 0.300 | 0.413 | 0.006 |
| | +Bayes-Q | 0.450 | 0.478 | 0 |
| | +Bayes-M | 0.550 | 0.436 | 0 |
| | +Bayes-QM | 0.724 | 0.478 | 0 |
| GPT-4o | LM | 0.617 | 0.454 | 0 |
| | +Bayes-Q | 0.729 | 0.512 | 0 |
| | +Bayes-M | 0.667 | 0.455 | 0 |
| | +Bayes-QM | 0.900 | 0.506 | 0 |

Table 9: *Guess Who?* QA: aggregated success rates and breakdown by model and strategy.

success rate increases from 0.300 to 0.724. As with the Battleship experiments, we can therefore see that Bayesian strategies uplift weak LMs past the performance of much stronger pure LM agents.

The middle conditions *Bayes-Q* and *Bayes-M* also separately improve as compared to both base conditions, suggesting that neither base LM optimally integrates information from all previous questions into its guessing strategy (otherwise we'd see no improvement through *Bayes-M*), and that neither model is reliably able to ask the optimal question at any given point during the game.

The *Bayes-Q* and *Bayes-QM* conditions also significantly raise EIG ($0.454 \rightarrow 0.506/0.512$ for GPT-4o, and $0.413 \rightarrow 0.478/0.478$ for Llama-4-Scout). Redundant (EIG = 0) questions are already rare – GPT-4o never asks any such question in any condition, Llama-4-Scout only asks then 0.6% of the time in the base LM condition – but no such questions are asked in any condition that optimizes for EIG. This is encouraging, since it means our EIG-based reranking approach is finding better questions than the first question the LM considers.

In general, these results point to the generality of this framework, having brought substantial improvements to the performance of LM agents in two very different information-seeking domains.

### E.6 *Guess Who?* PROMPTS

#### E.6.1 SYSTEM PROMPT

---

**System**

```
You are playing Guess Who. The other player has selected a character from the list
below, and your goal is to guess which character they have selected by asking
yes-or-no questions about the character's traits.

You can ask 8 questions in total, and you have to make your guess after asking all
of them. If you guess correctly, you win the game. If you guess incorrectly, you
lose the game.

Given this game history:

<history>
{history}
</history>

This is the list of characters you can ask questions about:

<characters>
{characters}
</characters>
```

---

This prompt is referred to as "{context}" when mentioned in other prompts.

### E.6.2 PLAYER PROMPTS

**Guess**

```
{{context}}

Your task is to make your one and only guess about the character. Make sure you
consider the context of the theme and all previous questions and answers.

Guess from the list above.

Please think about your answer step by step. When you have come up with a final
answer, respond with your guess wrapped in <answer></answer> tags, and optionally
square brackets, e.g. <answer>Mike</answer> or <answer>[Mike]</answer>
```

**Non-EIG Question**

```
{{context}}

Your task is to ask a single question that will help you gain the most information
possible about the secret word. You can ask any question, but is must be answerable
with a Boolean answer (yes/no). Make sure your questions are clear, specific and
different from ones you have asked previously, to avoid pigeonholing a potentially
wrong answer too quickly.

You have {{remaining_questions}} questions left, including this one.

Please think about your answer step by step. When you have come up with your
question, please wrap it in <answer></answer> tags: e.g. <answer>Is the character
male?</answer>
```

**EIG Question**

```
{{context}}

Your task is to generate {{k}} question(s) that will help you gain the most
information possible about the secret word. Each question must be answerable with a
Boolean answer (yes/no).

You have {{remaining_questions}} batches of {{k}} question(s) left, including this
one.

Please think about your answer step by step. When you have come up with your
question, please return your question(s) as a JSON dictionary with numbered keys,
wrapped in <answer></answer> tags like this: <answer>{{"1": "Is the character
male?", "2": "Do they have a mustache?", "3": "Do they have a hat?"}}</answer>

IMPORTANT: Use proper JSON format with double quotes around both keys and values.
```

### E.6.3 GAME/SAMPLER PROMPTS

**Consistency**

```
You are playing Guess Who. Your task is to determine which characters from the list
below are consistent with the constraint implied by the question.
```

```
The most recent question is "{{question}}".

Given the following characters:

<characters>
{{characters}}
</characters>

Respond with a JSON dictionary wrapped in <answer></answer> tags where the keys are
the characters and the values are either "yes", if the character satisfies the
constraint given in the question, or "no" if they do not.

<answer>{{"character1": "yes", "character2": "no", "character3": "yes"}}</answer>

IMPORTANT: Use proper JSON format with double quotes around both keys and values.
```

