# OpenReview forum: "Shoot First, Ask Questions Later? Building Rational Agents that Explore and Act Like People"
_ICLR.cc/2026/Conference — ICLR 2026 Oral_

### Official Review · Reviewer_QbJw · 2025-10-23

**Soundness:** 4
**Presentation:** 4
**Contribution:** 3
**Rating:** 8
**Confidence:** 5

**Summary:**

The present paper evaluates and improves the ability of LLMs to ask goal-directed questions and take actions in a dynamic environment. For this, the authors design a novel task called Collaborative Battleship. They run a human study on this task and compare human performance to that of various LLMs. This revealed a performance gap that the authors then subsequently address by developing Bayesian-inspired inference-time strategies for LLMs, leading to significant improvement in LLM performance.

**Strengths:**

I enjoyed this paper a lot. It has all the ingredients for a great paper: a new task, human evaluation, a decent set of different models that are evaluated, a novel technique for improving models, and demonstration that the findings generalize to another domain. The paper was exceptionally well written and easy to follow. The method is clean and simple, yet effective. CaptainQA is an interesting agentic test bed for LLMs and the whole methodology fits thematically well into ICLR.

**Weaknesses:**

I found the usage of the term Bayes-rational weird. Best to my knowledge, this is not an accepted term in the literature. It implies that the strategies developed by the authors are Bayes-optimal, which is not the case (as also noted by the authors). To avoid this confusion, I would suggest using a different term instead.

There is not so much negative to say about the paper. Perhaps the only downside is that, while the results and methods are interesting, they are not groundbreaking. For me, that is the only reason for giving this paper a score of 8 (instead of the full 10).

Minor:
* RSA not defined (p9).

**Questions:**

The indicator function in Equation 1 seems to be redundant (unless I am missing something).

The authors find that GPT-5 does not significantly benefit from Bayesian question or move selection. This raises the question of whether the proposed is useful for future models. Do you think that there is a risk of this being the case?

I was a bit confused by the description of Equation 7. Why is \pi_{t+1}^a introduced but then never used? Why p_{t+1}^{hit} defined as a distribution over questions? That seems strange. Why is u_t^* mentioned twice under step 2?

---

> ### Author Response · Authors · 2025-11-19
> **Re: Detailed review, helpful refinements to math presentation**
>
> Hello! Thank you so much for the positive review! We’re so delighted that you found the paper enjoyable and easy to follow. We’re very excited about this work and we’re glad that you feel they fit well thematically into ICLR.
>
> We really appreciate your attention to detail, especially regarding the mathematical presentation (see discussion below). Let’s start with some of the minor points and math before moving onto the broader questions.
>
> ### **Minor points**
>
> *“Bayes-rational” implies Bayes-optimal; this term is not standard in the literature / could be confusing*
>
> Thanks for pointing this out. “Bayes-rational” is intended as a gesture towards existing accounts of Bayesian rationality (e.g., Anderson, 1990; Oaksford & Chater, 1999, 2007; etc.). That said, your point is well-taken that this term isn’t especially common in the literature and could be ambiguous.
>
> We’ve now substituted this term for “resource rational” (e.g., Leider & Griffiths, 2020; Leider, Callaway, and Griffiths, 2025; Icard, 2025), which is both more contemporary and self-explanatory. Thanks again for the suggestion — please see the updated PDF and let us know if this sounds amenable to you.
>
> *RSA not defined (p9)*
>
> Thanks, we’ve updated the relevant line to explicitly state “rational speech acts (RSA)”.
>
> ### **Mathematical presentation**
>
> *The indicator function in Equation 1 seems to be redundant (unless I am missing something).*
>
> Eq. 1 concerns the predictive probability that the true answer will be “yes” and is defined as follows:
>
> $$p\_t = P(A\_t = 1 \mid x, \\mathcal{H}\_{1:t}) = \\sum\_{s \in \\mathcal{S}\_{\vdash x}}\\pi\_t(s) \\mathbf{1}\[ f\_{q\_t}(s) = 1 \]$$
>
> The indicator function here is intentional: it ensures that we only marginalize over the probability mass $\pi_t(s)$ of boards $s$ for which the question returns “True.”
>
> *Questions about Eq. 7:*
>
> These are really helpful questions — thanks so much for taking the time to review the math carefully. We’ve tweaked Eq. 7 and the surrounding section for clarity based on the points you raised. Please take a look at the updated PDF and let us know if you think there’s anything else than needs clarification.
>
> *Why is* $\pi_{t+1}^a$ *introduced but then never used?*
>
> This term was originally introduced to support math that got moved to the appendix. We agree that it’s no longer necessary to explicitly define the post-answer belief here so we’ve removed the term.
>
> *Why* $p_{t+1}^{hit}$ *defined as a distribution over questions?*
>
> Good catch — this should be defined as a distribution over tile locations $u$ so as to remain consistent with Eq. 6. We’ve updated Eq. 7 accordingly. (Note that this also means the inner maximization over $u$ unrevealed is unnecessary.)
>
> *Why is* $u_t^\*$ *mentioned twice under step 2?*
>
> This was a typo; it should be $u_t^\*$ and $u_{t+1}^\*$. Thanks for the catch!

---

> > ### Author Response · Authors · 2025-11-19
> > **Re: Broader points**
> >
> > *GPT-5 does not significantly benefit from Bayesian question or move selection.. is there a risk that this means the proposed methods won’t be useful for future models?*
> >
> > While this might be a concern if we only care about task performance, in thinking about future models, an increasingly important part of the picture is *efficiency*. In a very short amount of time, LLMs (and especially reasoning models) have started to consume a significant and growing share of global energy resources. Our findings show that we can achieve ~100x cost reduction on our task by using Bayesian strategies in conjunction with smaller models. Another way of viewing this result is that, even if frontier models like GPT-5 have reached human-level performance on tasks like ours, there is still significant room for improvement in terms of efficiency.
> >
> > *While the results and methods are interesting, they are not groundbreaking. For me, that is the only reason for giving this paper a score of 8 (instead of the full 10).*
> >
> > We're delighted to hear that you considered giving the paper a 10! It's up to you, but since this is a rebuttal, let us make a brief pitch for why you might consider increasing your score.
> >
> > First and foremost, in your own words, you enjoyed reading the paper and feel that it has all the ingredients for a great CS paper. It’s no secret that research quality and rigor is a growing issue at the major AI conferences. In a field increasingly focused on rapid publication cycles, we invested significant time and care piloting our human study and developing theoretical models that were both interesting and generalizable. To the extent that you found our approach "clean and simple," that reflects considerable effort to distill complex ideas into their clearest form. (Thank you as well for your valuable contributions to the theoretical presentation.)
> >
> > Regarding "groundbreaking" contributions: we believe this paper fundamentally challenges how the field currently thinks about scaling AI. In an era where frontier models consume increasingly unsustainable resources, demonstrating that weaker models with Bayesian strategies can outperform GPT-5 at orders of magnitude lower cost is likely to turn some heads and spark productive discussions. To further solidify this argument, we’re currently running experiments on a third task domain (also from TextArena), which we aim to add in a subsequent revision.
> >
> > As authors and reviewers, we have the opportunity to define what the community considers an "outstanding" paper that ought to be read by a broad audience. If this work fits the bill for you, we'd be really grateful for your support. Thank you again for your time and consideration.

---

> > > ### Comment · Reviewer_QbJw · 2025-11-24
> > >
> > > Thanks for the response! Happy with the answers and minor changes. It's a great paper that should be definitely accepted.

---

### Official Review · Reviewer_g4p6 · 2025-10-29

**Soundness:** 3
**Presentation:** 3
**Contribution:** 2
**Rating:** 4
**Confidence:** 3

**Summary:**

This paper introduces Collaborative Battleship, a novel two-player task designed to evaluate the trade-offs between information-seeking (asking questions) and exploitation (taking actions) in language model (LM) agents under communication constraints (yes/no answers). The authors collected a human dataset, BATTLESHIPQA (N=42 participants, 126 games), establishing human baselines for question quality and strategic play. They compare human performance to various LMs, finding that while weaker LMs struggle, frontier reasoning models approach human levels. To improve agent rationality, the paper proposes inference-time strategies based on Bayesian Experimental Design (BED), using Sequential Monte Carlo (SMC) to approximate belief states and guide question selection (maximizing EIG), action selection (maximizing hit probability), and the explore/exploit decision (one-step lookahead). These strategies significantly boost performance, enabling weaker LMs (Llama-4-Scout) augmented with Bayesian methods to outperform average humans and stronger LMs like GPT-5 in win rate, at substantially lower computational cost. Findings were replicated on the Guess Who? game. The authors provide IRB approval details and plan to release the dataset and code.

**Strengths:**

**Novel Task and Dataset:** Collaborative Battleship provides a clean, interpretable environment for studying the explore/exploit dilemma and grounded communication. The BATTLESHIPQA dataset, collected from human interactions, is a valuable contribution for benchmarking and analysis.

**Principled Bayesian Framework:** The application of BED principles (EIG maximization, belief updating via SMC, MAP action selection) provides a strong theoretical grounding for the proposed inference-time strategies.

**Strong Empirical Results:** The Bayesian strategies demonstrate significant improvements in LM agent performance across multiple metrics (accuracy, EIG, F1, win rate). The finding that augmenting weaker LMs can lead to super-human and SOTA-LM-beating performance at lower cost is particularly compelling.

**Generalization:** Successful replication of performance gains on the Guess Who? task suggests the framework's potential applicability to other information-seeking domains.

**Transparency and Reproducibility:** The paper includes ethical considerations for human subjects, clear plans for releasing code and data, and explicit disclosure of AI assistance.

**Weaknesses:**

**Limited Scope:** While the paper introduces an interesting method, its evaluation is confined to a specific domain defined by the authors, with generalization demonstrated only on one additional ad-hoc task. This limited scope makes it difficult to assess the method's potential for broader applicability and overall impact.

**"Surpass Humans" Claim Qualification:** The claim that augmented LMs surpass human performance needs stronger qualification. Details on human participants' prior experience, the number of trials per condition, and how performance compares when normalizing for computational resources (cost, latency, tokens) are necessary for a fair comparison. The large cost disparity is noted but not integrated into the primary win-rate comparisons.

**Rationality Framing:** The paper frames the goal as building "rational" agents using Bayes-optimal principles (BED). However, human behavior often follows boundedly rational heuristics. The evaluation primarily uses game performance (win rate, F1) as a proxy for rationality, potentially conflating task success with optimal information processing under constraints.

**Potential Scaffolding Effects:** The interaction between different components (question generation, answering, strategy selection, potential code generation for grounding) needs further ablation. It's unclear if performance gains could stem from implicit prompt leakage or interactions between modules rather than purely the Bayesian logic.

**Questions:**

1. Could you provide more details on the human study participants? What was their prior experience with Battleship? How were the number of human trials balanced against LM evaluations in terms of total interaction opportunities or budget?

2. Could you please provide performance results (e.g., win rate, F1 score) normalized by computational cost (tokens, latency, API cost)?  Cost-controlled comparisons (e.g., win rate vs. budget curves) would strengthen the claims about efficiency.

3. Can you provide ablations that isolate the capabilities of the Captain (questioner/actor) and Spotter (answerer) roles? For instance, how does a strong Captain perform with a weak Spotter, and vice-versa? How much does the specific code generation strategy contribute to the Spotter's grounding?

4. How sensitive are the results of the Bayesian strategies to hyperparameters like the SMC particle count, the number of candidate questions sampled (k for $Q_{Bayes}$), the decision discount factor ($\gamma$ for $D_{Bayes}$), and decoding parameters (e.g., temperature)?

5. What was the inter-rater reliability for the manual annotation of human questions and gold answers in BATTLESHIPQA?

6. Consider citing and comparing with SPIN-Bench (Yao et al., 2025) which evaluates LLMs in multi-agent cooperative and strategic settings (like Hanabi)  and documents coordination failures, making it relevant context for evaluating interactive agent strategies.

---

> ### Author Response · Authors · 2025-11-19
> **Re: Points raised in your review**
>
> Thank you for the review. Based on your comments, we’ve provided an additional **cost-weighted win-rate analysis** and are also in the process of running experiments on a potential additional task domain (see below).
>
> ### **Points addressed in existing submission**
>
> While the review raises a number of points, many of these are already addressed in the submission. Before turning to new analysis, we reiterate/clarify these below:
>
> *Could you provide more details on the human study participants? What was their prior experience with Battleship?*
>
> Full details on the human study, including participant demographics, filtering criteria, and trials per condition, are provided in the **existing A.2**.
>
> The human study included a tutorial to familiarize participants with the rules of the game; we did not assume prior experience with Battleship. However, we did set specific criteria on Prolific to exclude participants who had previously completed any prior study (e.g., pilot studies) from our group that specifically used the Collaborative Battleship protocol. This point has been clarified in the revised PDF (see A.2).
>
> *What was the inter-rater reliability for the manual annotation of human questions and gold answers in BATTLESHIPQA?*
>
> Inter-annotator agreement ranged from 94.0–99.6%; details on IRR are provided in the **existing A.3 (see Fig. 7a)**.
>
> *Can you provide ablations that isolate the capabilities of the Captain (questioner/actor) and Spotter (answerer) roles?*
>
> An extensive set of ablations is **already included in the paper (Sections 4.2-4.3)**:
>
> - [S4.2] Effect of code generation (Base, CoT, Code, CoT+Code)
> - [S4.3] Interactions between question, move, and decision components (see Table 1):
>     - LM+Bayes-Q: Only Bayesian question-generation
>     - LM+Bayes-M: Only Bayesian move selection
>     - LM+Bayes-QM: Bayesian question-generation + move selection
>     - LM+Bayes-QMD: Bayesian question-generation + move selection + decision function (one-step lookahead)
>
> These ablation conditions were explicitly designed to isolate the different components of the Captain and Spotter models, including the specific aspects referenced by the reviewer (question generation, answering, strategy selection, potential code generation for grounding).
>
> *It's unclear if performance gains could stem from implicit prompt leakage or interactions between modules rather than purely the Bayesian logic.*
>
> We certify that there is no unintentional “leakage” of information between modules in our implementation. The prompt text associated with each module is provided in the Prompt Library (Appendix C). We are happy to provide an anonymized version of the code if this is something you wish to verify for yourself; otherwise, the code will be available on publication.
>
> Regarding “interactions between modules,” we carefully chose the above set of ablation conditions to isolate the relative contributions of each module. This analysis **(Fig. 4a and Tables 4-5)** allows us to confirm that the full model (LM+Bayes-QMD) outperforms the ablations.
>
> *The paper frames the goal as building "rational" agents using Bayes-optimal principles (BED). However, human behavior often follows boundedly rational heuristics.*
>
> The difference between optimal and bounded notions of rationality is **already a central part of our paper’s framing**; e.g., L186-188 (*“These strategies are not necessarily globally optimal, but rather Bayes-rational…”*); L444-451 (*”Our work is broadly informed by “resource rational” accounts that describe how human behavior is shaped by cognitive constraints…”*). Our approach models *boundedly rational* human behavior via heuristics like MAP inference and one-step lookahead, as opposed to globally-optimal strategies (e.g., long-horizon planning).
>
> As noted by another reviewer (QbJw), the term “Bayes-rational” is not as widely-established in the literature as we’d believed and could lead to some confusion. For this reason, we’ve substituted this term for “resource rational”, which we hope will help to clarify this point. Please see the response to Reviewer QbJw for more discussion.
>
> *The evaluation primarily uses game performance (win rate, F1) as a proxy for rationality.*
>
> In total, we examine 5 different metrics (F1, move count, win rate, questions asked, EIG); only two of these (F1 and win rate) are end-to-end task success metrics. Our secondary analysis on question EIG (Fig. 4b) and question timing (Fig. 4c), as well as an extensive set of qualitative examples (A.4 and D.2, Tables 6-7), provide a holistic picture of human and agent behavior beyond task success.

---

> > ### Author Response · Authors · 2025-11-19
> > **Cost-weighted win rate analysis**
> >
> > *The claim that augmented LMs surpass human performance needs stronger qualification… normalizing for computational resources (cost, latency, tokens) are necessary for a fair comparison. The large cost disparity is noted but not integrated into the primary win-rate comparisons.*
> >
> > **Tl;dr:** At the reviewer’s request, we provide a cost-normalized win-rate comparison below. We find that (1) weighting for cost *does not* change our findings (small models can be made to outperform large models); (2) however, it does accentuate the advantage of smaller models to a degree that is potentially misleading. For instance, adjusting for cost, the baseline Llama-4-Scout (with no Bayesian enhancements) jumps from 0% → 99.2% win rate over GPT-5, despite never winning a single game. Overall, we find that weighting for cost introduces extra technical complexity while skewing the results in way that is likely to affect interpretability for many readers. We unpack these concerns further below.
> >
> > **Full response:** The existing comparisons in our paper are based on measures of task performance, such as the win-rate metric cited by the reviewer. Across various experiments, we observe that both our method—as well as strong frontier models—surpass mean human performance (e.g., Claude-4 on the Spotter task; GPT-5 on both Spotter and Captain tasks). These claims are based on straightforward comparisons of the metrics (e.g., win-rate percentage) across conditions with no further adjustment.
> >
> > The reviewer suggests that accounting for cost is necessary to provide “fair comparison” both between models vs. humans and between models. While this sounds superficially like a reasonable request, in practice, weighting the primary win-rate comparisons by cost poses a number of issues.
> >
> > 1. It is not clear how to construct a fair cost-weighted win rate comparison between models and humans. While there is a “cost” associated with running human experiments, participant pay is based on minimum hourly wage laws and variable bonuses. These factors are not directly comparable to LLM inference cost, which is based on token counts and API pricing set by model providers.
> > 2. As noted by the reviewer, there is a large cost disparity between the smallest model (Llama-4-Scout) and the largest model (GPT-5); it is unclear how this 2 orders of magnitude difference in cost can be incorporated while maintaining informative signal from the win rates.
> >
> > To avoid these issues, we (1) restrict to model-model comparisons, and (2) re-normalize the existing F1-based win rates by the respective inference cost of each LLM (Table 5).
> >
> > Below, we provide a table summarizing how the win rate shifts when adjusting for cost. Note the substantial and misleading shift in apparent performance: **Llama-4-Scout (LM) jumps from 0% → 99.2% win rate over GPT-5 (LM), despite never winning a single game.**
> >
> > We hope that this addresses the reviewer’s request, while illustrating why we believe it is clearer and simpler to report win rates (Fig. 18) and compute cost (Table 5) separately. Please let us know if you agree with these conclusions; otherwise, we are happy to provide further clarification.
> >
> > | Competitor A | Competitor B | Original Win Rate | Cost-Weighted Win Rate | Cost Ratio |
> > | --- | --- | --- | --- | --- |
> > | **Llama-4-Scout LM** | **GPT-5 LM** | **0** | **0.992** | **119.858** |
> > | Llama-4-Scout LM | GPT-5 +Bayes-Q | 0 | 0.997 | 306.867 |
> > | Llama-4-Scout LM | GPT-5 +Bayes-M | 0 | 0.986 | 69.142 |
> > | Llama-4-Scout LM | GPT-5 +Bayes-QM | 0 | 0.996 | 232.1 |
> > | **Llama-4-Scout LM** | **GPT-4o LM** | **0.253** | **0.967** | **29.217** |
> > | Llama-4-Scout LM | GPT-4o +Bayes-Q | 0.148 | 0.985 | 67.908 |
> > | Llama-4-Scout LM | GPT-4o +Bayes-M | 0 | 0.94 | 15.617 |
> > | Llama-4-Scout LM | GPT-4o +Bayes-QM | 0.006 | 0.978 | 44.175 |
> > | Llama-4-Scout LM | GPT-4o +Bayes-QMD | 0 | 0.977 | 42.35 |
> > | ... |  |  |  |  |
> > | **GPT-4o LM** | **GPT-5 LM** | **0.043** | **0.804** | **4.102** |
> > | GPT-4o LM | GPT-5 +Bayes-Q | 0.062 | 0.913 | 10.503 |
> > | GPT-4o LM | GPT-5 +Bayes-M | 0.065 | 0.703 | 2.367 |
> > | GPT-4o LM | GPT-5 +Bayes-QM | 0.043 | 0.888 | 7.944 |
> > | ... |  |  |  |  |

---

> > > ### Author Response · Authors · 2025-11-19
> > > **Re: Miscellaneous concerns**
> > >
> > > *How sensitive are the results of the Bayesian strategies to hyperparameters (e.g., SMC particle count, candidate question k, discount factor* $\gamma$*, temperature)?*
> > >
> > > In many cases, the behavior of hyperparameters can be approximately characterized through our existing analysis. For instance, in Fig. 4b, we can see the effect of scaling k on the question EIG, which tracks with downstream task performance. In other cases, we can reason about the effects of hyperparameters based on their function (e.g., decreasing the discount factor $\gamma$ will lead to more myopic behavior; increasing temperature will lead to more diversity in question sampling).
> > >
> > > Unfortunately, we aren’t able to run extensive hyperparameter sweeps in the end-to-end setting as the experiments are quite costly (see Table 5). That said, the results we present (with fixed hyperparameters) in general represent a *lower bound* on the performance of our Bayesian strategies with full hyperparameter tuning.
> > >
> > > *The evaluation is confined to a specific domain defined by the authors (Battleship), with generalization demonstrated only to one additional task (Guess Who?), limiting potential broader applicability and impact.*
> > >
> > > The core contribution of this paper is the introduction of a novel task and dataset (Collaborative Battleship) and an in-depth evaluation of both humans and agents on this domain. While we believe this already qualifies as a significant research contribution, we reproduce our evaluations on Guess Who? task from TextArena (Guertler et al., 2025) to illustrate that our methods generalize to a task setting that we did not define.
> > >
> > > While we don’t view the point of this paper as a benchmarking exercise, in response to your feedback, we’ve secured an additional $2.5K in API credits and are now running experiments on a third task domain (also from TextArena). It’s unclear at this point that these will be ready in time for ICLR rebuttal, but we will follow up in the event that we have additional results for you to review.
> > >
> > > Looking ahead, future work could certainly seek to extend our methods to other tasks, such as the SPIN-Bench (Yao et al., 2025) benchmark you highlighted—however, we would stress that we view this as very much an extension beyond the scope of a single paper.
> > >
> > > *Consider citing and comparing with SPIN-Bench (Yao et al., 2025) which evaluates LLMs in multi-agent cooperative and strategic settings (like Hanabi)*
> > >
> > > Thank you for the suggestion — this seems like interesting related work. We’ve included a reference to this work in the Introduction (S.1); please see the updated PDF.

---

> > > > ### Comment · Reviewer_g4p6 · 2025-11-19
> > > > **Thank you for the nice and detailed rebuttal. I will increase my score.**
> > > >
> > > > Thank you to the authors for the detailed and convincing rebuttal. I do appreciate the clarification regarding the human study details in Appendix A.2/A.3. Regarding the cost-weighted analysis, the counter-example of Llama-4-Scout effectively addresses the concern and justifies that keeping cost and win-rate separate is reasonable. In addition, I really like the adoption of 'resource rational' framing. Regarding extension to other domains, I fully agree with your words that " this already qualifies as a significant research contribution", and I believe there is some misunderstanding here -- I am not expecting more experiments on other domains which cost too much unnecessary time and money. Instead, I am looking for some justification or intuition on how you expect the current results will generalize to other domains and make a larger impact. In response to your detailed and convincing rebuttal, I have increased my score, and still, I do hope to see a short paragraph demonstrating the potential impact of the methods and the dataset mentioned in the paper (e.g. why "BattleShip", why you believe it is the best for the goal, why other games/settings don't qualify, and how you interpret and transfer the results for insights in a more general domain).

---

> ### Author Response · Authors · 2025-11-19
> **Re: score increase -- thank you!**
>
> Thank you very much for the thoughtful follow-up and for increasing your score. We really appreciate your prompt response to the rebuttal and we’re grateful for vote in favor of acceptance. To your request for a clearer explanation of why we believe Battleship is the right setting, and how the results generalize, here is a brief statement:
>
> Battleship is intentionally designed as a model setting for Bayesian experimental design (BED): it offers (1) a grounded, fully-specified latent state, (2) a combinatorially large hypothesis space, (3) a well-defined notion of “informative” vs. “uninformative” questions, and (4) an infinite question space *expressed as natural language*. Many other game environments are less suitable for this kind of analysis because they lack one or more of these components. For instance, other strategic games like Hanabi or Mastermind incorporate information-seeking, but the query set is finite/small (e.g., only 10 total colors+numbers in Hanabi) and hints/queries aren’t expressible in natural language. Battleship, in contrast, gives us a controlled environment where natural language questions have programmatic semantics and can be evaluated in terms of expected information gain—similar in spirit to how the ARC Challenge provides a structured substrate for evaluating pattern induction, but here with questions as the object of study.
>
> This structure is precisely what enables the broader impact: the phenomena we observe in Battleship, such as how models trade off exploration vs. action, arise in many real-world “needle-in-a-haystack” settings where information is costly. For example, in clinical reasoning, physicians ask targeted follow-up questions to refine a hypothesis about a patient’s condition; recent works like MediQ [1] formalize this as a question-asking problem, and our methods can be applied there nearly directly. Similar dynamics also appear in settings like pharmaceutical drug discovery, where experiments are expensive and “hits” (e.g., molecular compounds that achieve desired potency against some biological target) are rare [2]. Battleship gives us a clean testbed to analyze these behaviors quantitatively, but the underlying principles transfer to any domain that involves sequential, hypothesis-driven information gathering under uncertainty.
>
> We hope this response satisfies your questions about impact/generalization. Thank you again for your engagement here!
>
> [1] Li, S.S., Balachandran, V., Feng, S., Ilgen, J., Pierson, E., Koh, P.W., & Tsvetkov, Y. (2024). MediQ: Question-Asking LLMs and a Benchmark for Reliable Interactive Clinical Reasoning. Advances in Neural Information Processing Systems 37.
>
> [2] Wang, H., Fu, T., Du, Y. et al. Scientific discovery in the age of artificial intelligence. Nature 620, 47–60 (2023). https://doi.org/10.1038/s41586-023-06221-2

---

> > ### Comment · Reviewer_g4p6 · 2025-11-20
> > **Thanks for the explanation and insights**
> >
> > Thank you for the detailed explanation. That makes sense to me.

---

### Official Review · Reviewer_VQ6M · 2025-11-01

**Soundness:** 4
**Presentation:** 3
**Contribution:** 3
**Rating:** 8
**Confidence:** 3

**Summary:**

Paper introduces the collaborative battleship game, where a “captain” and a “spotter” collaborate in the game of battleship. The captain must decide between asking the spotter a question, or taking a hit at a board position, thereby having to balance information seeking and reward seeking.

The authors collected the BATTLESHIPQA human dataset where pairs of humans collaborated to play the game (42 participants, 21 pairs). This establishes human performance, the kind of questions humans ask, and how accurately the human (spotter) responds.

They then evaluated 15 LMs in the role of the spotter (i.e. Q&A), finding that the best frontier models can do well, but they show greater degradation in hard (i.e. context dependent) questions. Then, they evaluate 3 LMs in the role of the captain, with various “bayes rational” strategies.  The “bayes rational” strategies improve the performance of weak LMs to the level of / slightly beyond GPT-5.

Finally, they extended the bayesian inference framework to other information seeking games and see some performance gains.

**Strengths:**

There are numerous strengths with this paper. To start, the collaborative battleship game sets up an interesting scenario for the captain agent, where they not only have to balance information seeking vs. reward seeking behaviour, but also maintain uncertainty about the correctness of its collaborator (noisy spotter agent). Information seeking in the space of natural language is an interesting setting to study.

The human dataset provides valuable information to ground “human level” performance, the type of questions asked, and human-like behaviour. The LM evaluations are comprehensive, with interesting findings including code generation boosting spotter accuracy. The “bayes rational” captain strategies improving worse models to perform at GPT-5 level has useful application implications.

Overall, the paper is very well written and presented.

**Weaknesses:**

A number of simplifying assumptions were made in the “bayes rational” modelling choices. For instance, modelling with fixed $\epsilon$ (as the authors already point out) and $\gamma$ (not sure how this is set), and only modelling single-step look-ahead are simplifying assumptions. I do not think this detracts from the main point of the paper, as to my understanding this paper is about improving empirical performance of weak LMs using some cognitively inspired “bayes rational" strategies. Nevertheless, it is worth pointing this out for scientific rigour.

However, there are a few simplifications that I am quite confused about, and likely warrant deeper discussions. I ask them in the question section below.

**Questions:**

### On assumptions

**(1)**
is maximizing EIG (Eq 5, L214) in this game the “right” thing to do to maximize performance of hitting a ship? My understanding is that in some games from previous works (e.g. ActiveACRE, blickets, feature world) [1,2,3], the explicit goal is to find out how the environment works. In these cases, directly optimizing for information gain is the “right” thing to do.

In battleships, seeking information is only ever in service of the goal of hitting ships. Therefore, rather than computing the post-question hit probability (Eq 7) with the max-EIG question (Eq 5), i.e. $p_{t+1}^{hit} (q_{t}^{*} | x, H)$, shouldn’t we directly find $\arg\max_q p_{t+1}^{hit}(q | x, H)$ instead, and use that in the decision rule in L217? Is there no scenario in which this will change which question is selected? This is discussed empirically in L405-412 but it would be good to discuss a bit theoretically as well.

**(2)**
I do not think the information gained from the “shoot” action is appropriately considered. The act of shooting perfectly reveals information about whether or not a tile contains a ship, akin to asking a noise-less spotter “is there a ship on this tile”. Indeed, for a 8x8 board and a budget of 40 shots, 40/64 = 62.5% of tiles can be revealed this way. Thus, shouldn’t the captain’s decision really be between (i) asking a question based on post-hit probability (per point (1) above), and (ii) choosing a shot that will both hit a current target and/or increase post-hit probability of the next shot?


### Minor clarifications

1. What does Figure 3 b and c error bars denote? Please label / state in figure caption
2. How is the discount factor $\gamma$ in L216 selected?

---

[1] Piriyakulkij, Top, et al. "Doing experiments and revising rules with natural language and probabilistic reasoning." Advances in Neural Information Processing Systems 37 (2024): 53102-53137.

[2] GX-Chen, Anthony, et al. "Language Agents Mirror Human Causal Reasoning Biases. How Can We Help Them Think Like Scientists?." arXiv preprint arXiv:2505.09614 (2025).

[3] Sawyer, Danny P., et al. "Can foundation models actively gather information in interactive environments to test hypotheses?." arXiv preprint arXiv:2412.06438 (2024).

---

> ### Author Response · Authors · 2025-11-19
> **Re: positive feedback and fantastic questions!**
>
> Hi! Thanks so much for your thoughtful review. We’re glad to hear that you enjoyed reading the paper, found the setting interesting, and that you feel the methods have useful application implications.
>
> We appreciate your insightful questions, which raised several interesting points on the modeling side. Please see our answers below; we’ll add clarifications to the revision based on your feedback and any further discussion.
>
> Finally, thank you for highlighting a few related works in your review. These seem especially relevant and we’ve added [2, 3] to the paper ([1] was already cited). Please see the updated PDF.
>
> ### **On assumptions**
>
> **(1) Maximizing EIG vs. hit probability.** You’re correct that when selecting a question, there’s a choice between maximizing information about the board state (the current formulation) vs. maximizing hit probability.
>
> There are definitely scenarios in which this will change the question that is selected. For example, at the beginning of the game when the board is blank, the question “Are there more ships on the top half of the board than the bottom half?” has near-ceiling EIG because it perfectly partitions the space of boards into two (up to the boundary condition on $\gt$ vs. $\geq$ implied by “more”). However, this question doesn’t actually yield much information about any *particular* tile. Empirically, we find this question has $\Delta p^{\text{hit}}=0.006$, meaning it only marginally increases the probability of hitting a ship on the next move.
>
> As this example highlights, maximizing for EIG (Eq. 5) in general leads to more exploratory questions, whereas maximizing for $p^\text{hit}_{t+1}(q \mid \cdot)$ leads to more myopic questions. In practice, we find that the hit probability objective causes the Captain to favor confirmatory questions; e.g.,
>
> > Captain (LM): “Is there a ship at A6?”
> Spotter: Yes
> Captain: Fire at A6.
> >
>
> Interestingly, we also observe that humans occasionally ask these kinds of questions in our BattleshipQA dataset; e.g.,
>
> > Captain (Human 01JF…): “Since the questions don't cost anything confirming the final tile is C8”
> Spotter: Yes
> Captain: Fire at C8. (Game won)
> >
>
> That said, when it comes to building the models, maximizing for hit probability during question selection yields disproportionately conservative Captains that ask for confirmation with much higher frequency than humans.
>
> In sum, the above is why we model the question-asking strategy as max-EIG rather than max-hit. Both can be thought of as approximate heuristics for the optimal behavior (maximizing expected future hit probability **over the *full game horizon*). However, since this is intractable, we favor the max-EIG objective because it encodes a more exploratory strategy that better characterizes the behaviors we find in the human data.
>
> **(2) Modeling information gain from the “shoot” action.** This is a great point as well: shooting can be viewed as a form of information-gathering, akin to a noiseless version of the single-tile “confirmatory” questions highlighted above. The key difference is that, in our setting, questions are “free” (there’s no cost to asking a bad question aside from having wasted an opportunity to gain information). In contrast, every missed shot comes with a score penalty. For humans, misses also reduce the monetary bonus, which is presented as a salient element in the experiment UI.
>
> We designed the experiment in this way to model settings where the cost of gathering information (e.g., running additional medical diagnostic tests) is relatively low compared to the cost of taking actions (e.g., prescribing the wrong medication for a patient).
>
> That said, even with the penalty for misses, there may still be cases where it’s “worth it” to take a risk on shooting at a tile. For instance, if the Captain believes the Spotter is unreliable, then they may decide it’s necessary to just shoot and risk a miss. Another example is the situation where the Captain has run out of questions, then the only way to gain information about the board is by taking actions.
>
> In the most basic sense, our model *does* account for information gained from shooting because every action updates the posterior over boards. However, as you point out, the *decision function* specifically (”D” in Bayes-QMD) in the implementation that we presented does not. Accounting for this properly would require more explicitly trading off resource costs between shot and question budgets. One simple approach that we considered would be to extend the existing value of information (VoI) decision function to include additional terms that capture things like the cost of missing and the value of the remaining questions. However, these kinds of econometric-style models have a tendency to get overly complex / problem-specific -- we wanted to introduce theory that is general to a variety of info-seeking tasks, not just our Battleship scenario.

---

> > ### Author Response · Authors · 2025-11-19
> > **Re: Minor clarifications**
> >
> > 1. Thank you for the question. The error bars in Fig. 3 indicate 95% CIs; this has been noted in the caption (see updated PDF).
> > 2. We use $\gamma=0.95$, which was selected (somewhat heuristically) to encode a minor discount to the value of future action. (For other hyperparameters, we’re able to provide stronger theoretical justification; for instance, the noise parameter $\epsilon$ was calibrated to the empirical Spotter error in the benchmarks in S4.2.) That said, note that $\gamma$ is only used in the +Bayes-QMD condition. While it would’ve been nice to run more exhaustive hyperparameter tuning, the experiments Captain are quite costly (see Table 5). That said, the results we present (with fixed hyperparameters) in general represent a *lower bound* on the performance of our Bayesian strategies with full hyperparameter tuning.

---

### Meta-Review · Area_Chair_cYC5 · 2025-12-11

**Summary:**

Overall, the reviews were pretty positive about this paper. They praised the Collaborative Battleship task and accompanying human dataset as a novel and well-motivated testbed for evaluating strategic reasoning and exploration-exploitation trade-offs. They further highlighted the effectiveness of the proposed Bayesian inference strategies, which enabled weaker language models to achieve performance comparable to humans and state-of-the-art models.

The reviewers raised theoretical concerns regarding the design and framing of the Bayesian strategies. They questioned the accuracy of the term "Bayes-rational" given simplifying assumptions like single-step lookahead and fixed parameters. A core technical critique was whether maximizing Expected Information Gain (EIG) is the appropriate objective for a game where the goal is winning rather than pure system identification; specifically, reviewers noted that the current model may not properly account for the information gained directly from shooting actions. One reviewer also questioned the long-term utility of the method, observing that the strongest model evaluated (GPT-5) saw little benefit from the proposed inference-time strategies.

On the experimental side, the reviewers called for more rigorous benchmarking and ablations. The claim that the method allows models to surpass human performance was scrutinized, with requests for comparisons that normalize for computational costs, latency, and token usage to ensure a fair assessment against human constraints. Reviewers also asked for sensitivity analyses regarding hyperparameters such as particle counts and discount factors, as well as ablations to isolate the specific contributions of the Captain and Spotter roles. Finally, there were requests to clarify mathematical notation and figure details, such as the definitions of error bars.

**Reviewer Concerns:**

The authors:

- Explained their use of the term "Bayes-rational", but changed it to "resource rational" to avoid reader confusion.
- Provided a thorough explanation of the use of EIG as the objective.
- Gave a new cost-weighted win rate analysis to provide a fair assessment against humans.
- Noted how their technique could be used to make smaller models more performant (thus making efficiency a potential long-term relevance).
- Provided additional clarifications on math, figures, etc.

**Reviewer Scores:**

The initial scores were 8,4,8. The reviewer with a score of 4 explicitly said they would raise their score, and given that their concerns were largely addressed according to the AC's assessment, it is likely the new score would be 6. It's possible one of the other reviewer's would have raised their score as well, though one (QbJw) noted why they gave an 8, and the reasoning likely wouldn't have changed post rebuttal. The final scores would likely have been 8,6,8 or 9,6,8.

---

### Decision · Program_Chairs · 2026-01-26

Accept (Oral)